# Nonzero-sum Adversarial Hypothesis Testing Games

**Sarath Yasodharan**
Department of Electrical Communication Engineering
Indian Institute of Science
Bangalore 560 012, India
sarath@iisc.ac.in

**Patrick Loiseau**
Univ. Grenoble Alpes, Inria, CNRS, Grenoble INP, LIG & MPI-SWS
700 avenue Centrale
Domaine Universitaire
38400 St Martin d'Héres, France
patrick.loiseau@inria.fr

## Abstract

We study nonzero-sum hypothesis testing games that arise in the context of adversarial classification, in both the Bayesian as well as the Neyman-Pearson frameworks. We first show that these games admit mixed strategy Nash equilibria, and then we examine some interesting concentration phenomena of these equilibria. Our main results are on the exponential rates of convergence of classification errors at equilibrium, which are analogous to the well-known Chernoff-Stein lemma and Chernoff information that describe the error exponents in the classical binary hypothesis testing problem, but with parameters derived from the adversarial model. The results are validated through numerical experiments.

## 1 Introduction

Classification is a simple but important task that has numerous applications in a variety of domains such as computer vision or security. A traditional assumption that is used in the design of classification algorithms is that the input data is generated without knowledge of the classifier being used, hence the data distribution is independent of the classification algorithm. This assumption is no longer valid in the presence of an adversary, as an adversarial agent can learn the classifier and deliberately alter the data such that the classifier makes an error. This is the case in particular in security applications where the classifier's goal is to detect the presence of an adversary from the data it observes.

Adversarial classification has been studied in two main settings. The first focuses on adversarial versions of a standard classification task in machine learning, where the adversary attacks the classifier (defender/decision maker) by directly choosing vectors from a given set of data vectors; whereas the second focuses on adversarial hypothesis testing, where the adversary (attacker) gets to choose a distribution from a set of distributions and independent data samples are generated from this distribution. The main differences of the latter framework from the former are that: (i) the adversary only gets to choose a distribution (rather than the actual attack vector) and data is generated independently from this distribution, and (ii) the defender makes a decision only once after it observes a whole data sequence instead of making a decision for each individual data sample it receives. Both of these frameworks have applications in a variety of domains, but prior literature has mainly focused on the first setting; see Section 1.1 for a description of the related literature.

In this paper, we focus on the setting of adversarial hypothesis testing. To model the interaction between the attacker and defender, we formulate a nonzero-sum two-player game between the adversary and the classifier where the adversary picks a distribution from a given set of distributions, and data is generated independently from that distribution (a non-attacker always generates data from a fixed distribution). The defender on his side makes a decision based on observation of $n$ data points. Our model can also be viewed as a game-theoretic extension of the classical binary hypothesis testing problem where the distribution under the alternate hypothesis is chosen by an adversary. Based on our game model, we are then able to extend to the adversarial setting the main results of the classical hypothesis testing problem (see Section 2) on the form of the best decision rule and on the rates of decrease of classification errors. More specifically, our contributions can be summarized as follows:

1. We propose nonzero-sum games to model adversarial hypothesis testing problems in a flexible manner.
2. We show existence of mixed strategy Nash equilibria in which the defender employs certain likelihood ratio tests similar to that used in the classical binary hypothesis testing problem.
3. We show that the classification errors under all Nash equilibria for our hypothesis testing games decay exponentially fast in the number of data samples. We analytically obtain these error exponents, and it turns out that they are same as those arising in certain classical hypothesis testing problem, with parameters derived from the adversarial model.
4. We illustrate the results, in particular the importance of some assumptions, using simulations.

Throughout our analysis, an important difficulty lies in that the strategy spaces of both the players are uncountable; we believe, however, that it is an important feature of the model to be realistic.

## 1.1 Related Work

Adversarial classification and the security of machine learning have been studied extensively in the past decade, see e.g., [9, 20, 4, 14, 18, 22]; here we focus only on game-theoretic approaches to tackle the problem. Note that, besides the adversarial learning problem, game theory has been successfully used to tackle several other security problems such as allocation of monitoring resources to protect targets, see e.g., [8, 17]. We review here only papers relating to classification.

A number of game-theoretic models have appeared in the past decade to study the adversarial classification problem in the classical setting of classification tasks in machine learning. [9] studies the best response in an adversarial classification game, where the adversary is allowed to alter training data. A number of zero-sum game models were also proposed where the attacker is restricted on the amount of modifications he can do to the training set, see [16, 29, 28]. [6] studies the problem of choosing the best linear classifier in the presence of an adversary (a similar model is also studied in [7]) using a nonzero-sum game, and shows the existence of a unique pure strategy Nash equilibrium. Similar to our formulation, the strategy sets in this case are uncountable, and therefore showing the existence and uniqueness of Nash equilibrium needs some work. However, in our formulation, there may not always exist a Nash equilibrium in pure strategies, which makes the subsequent analysis of error exponents more difficult. [19] studies an adversarial classification game where the utilities of the players are defined by using ROC curves. The authors study Nash equilibria for their model and provide numerical discretization techniques to compute the equilibria. [12] studies a nonzero-sum adversarial classification game where the defender has no restriction on the classifier, but the attacker is limited to a finite set of vectors. The authors show that the defender can, at equilibrium, use only a small subset of "threshold classifiers" and characterize the equilibrium through linear programming techniques. In our model, the utility functions share similarities with that of [12], but we work in the hypothesis testing framework and with uncountable action sets, which completely modifies the analysis. Several studies appeared recently on "strategic classification", where the objective of the attacker(s) is to improve the classification outcome in his own direction, see [13, 11].

On the other hand, adversarial hypothesis testing has been studied by far fewer authors. [2] studies a source identification game in the presence of an adversary, where the classifier needs to distinguish between two source distributions $P_0$ and $P_1$ in which the adversary can corrupt samples from $P_0$ before it reaches the classifier. They show that the game has an asymptotic Nash equilibrium when the number of samples becomes large, and compute the error exponent associated with the false negative probability. [3] and [26] study further extensions of this framework.

A (non game-theoretic) hypothesis testing problem in an adversarial setting has been studied by [5], which is the closest to our work. Here, there are two sets of probability distributions and nature

outputs a fixed number of independent samples generated by using distributions from either one of these two sets. The goal of the classifier is to detect the true state of nature. The authors derive error exponents associated with the classification error, in both Bayesian and Neyman-Pearson frameworks using a worst-case maxmin analysis. Although we restrict to i.i.d. samples and let the non-attacker play a single distribution, we believe that our nonzero-sum game model with flexible utilities can better capture the interaction between adversary and classifier. There also exists extensive prior work within the statistics literature [15] on minimax hypothesis testing, which relates to our paper, but we defer a discussion of how our work differs from it to after we have exposed the details of our model.

Game-theoretic models were also used to study adversarial classification in a sequential setting, see [25, 1, 21], but with very different techniques and results.

## 2 Basic Setup and Hypothesis Testing Background

In this section, we present the basic setup results in classical binary hypothesis testing.

Throughout the paper, we consider an alphabet set $\mathcal{X}$ that we assume finite. In a classical hypothesis testing problem, we are given two distribution $p$ and $q$, and a realization of a sequence of independent and identically distributed random variables $X_1, \ldots, X_n$, which are distributed as either $p$ (under hypothesis $H_0$) or $q$ (under hypothesis $H_1$). Our goal is to distinguish between the two alternatives:

$$H_0 : X_1, X_2, \ldots, X_n \text{ i.i.d.} \sim p \quad \text{versus} \quad H_1 : X_1, X_2, \ldots, X_n \text{ i.i.d.} \sim q.$$

In this setting, we could make two possible types of errors: (i) we declare $H_1$, whereas the true state of nature is $H_0$ (Type I error, or false alarm), and (ii) we declare $H_0$ whereas the true state of nature is $H_1$ (Type II error, or missed detection). Note that one can make one of these errors arbitrarily small at the expense of the other by always declaring $H_0$ or $H_1$.

The trade-off between the two types of errors can be captured using two frameworks. If we have knowledge on the prior probabilities of the two hypotheses, then we can seek a decision rule that minimizes the average probability of error (this is the Bayesian framework). On the other hand, if we do not have any information on the prior probabilities, then we can fix $\varepsilon > 0$ and seek a decision rule that minimizes the Type II error among all decision rules whose Type I error is at most $\varepsilon$ (this is the Neyman-Pearson framework). In both of these frameworks, it can be shown that the optimal test is a likelihood ratio test, i.e., given $\mathbf{x}^n = (x_1, \ldots, x_n)$ we compute the likelihood ratio $\frac{q(\mathbf{x}^n)}{p(\mathbf{x}^n)}$ and compare it to a threshold to make a decision (with possible randomization at the boundary in the Neyman-Pearson framework). Here, $p(\mathbf{x}^n)$ (resp. $q(\mathbf{x}^n)$) denotes the probability of observing the $n$-length word $\mathbf{x}^n$ under the distribution $p$ (resp. $q$). See Section II.B and II.D in [23] for an introduction to hypothesis testing.

For large enough $n$, by the law of large numbers, the fraction of $i$ in an observation $\mathbf{x}^n$ is very close to $p(i)$ (resp. $q(i)$) under $H_0$ (resp. under $H_1$), for each $i \in \mathcal{X}$. Therefore, one anticipates that the probability of correct decision is very close to $1$ for large enough $n$. Hence, one can study the rate at which the errors go to $0$ as $n$ becomes large. It is shown that, under both frameworks, the error decays exponentially in $n$. In the Bayesian framework, the error exponent associated with the average probability of error is $-\Lambda_0^*(0)$, where $\Lambda_0^*(\cdot)$ is the Fenchel-Legendre transform of the log-moment generating function of the random variable $\frac{q(X)}{p(X)}$ under $H_0$, i.e., when $X \sim p$. In the Neyman-Pearson case, the error exponent associated with the Type II error is $-D(p||q)$ where $D$ is the relative entropy functional. The above error exponents are known as Chernoff information and Chernoff-Stein lemma, respectively (see Section 3.4 in [10] for the analysis on error exponents).

In this work, we propose extensions of the classical hypothesis testing framework to an adversarial scenario modeled as a game, both in the Bayesian and in the Neyman-Pearson frameworks; and we investigate how the corresponding results are modified. Due to space constraints, we present only the model and results for the Bayesian framework in the main body of the paper. The corresponding analysis for the Neyman-Pearson framework follows similar ideas and is relegated to Appendix A of the full version of this paper [27]. The proofs of all results presented in the paper (and in Appendix A of the the full version [27]) can be found in Appendix B of the full version [27].

# 3 Hypothesis Testing Game in the Bayesian Framework

In this section, we formulate a one-shot adversarial hypothesis testing game in the Bayesian framework, motivated by security problems where there might be an attacker who modifies the data distribution and a defender who tries to detect the presence of the attacker. Game theoretic modelling of such problems has found great success in understanding the behavior of the agents via equilibrium analysis in many applications, see Section 1.1. We first present the model and then elaborate on its motivations and on how it relates to related works in statistics.

## 3.1 Problem Formulation

Let $\mathcal{X} = \{0, 1, \ldots, d - 1\}$ denote the alphabet set with cardinality $d$, and let $M_1(\mathcal{X})$ denote the space of probability distributions on $\mathcal{X}$. Fix $n \geq 1$.

The game is played as follows. There are two players: the external agent and the defender. The external agent can either be a non-attacker or an attacker. In the Bayesian framework, we assume that the external agent is an attacker with probability $\theta$, and a non-attacker (normal user) with probability $1 - \theta$. The non-attacker is not strategic and she does not have any adversarial objective. If the external agent is a non-attacker, she generates $n$ samples independently from the distribution $p$. If the external agent is an attacker, she picks a distribution $q$ from a set of distributions $Q \subseteq M_1(\mathcal{X})$ and generates $n$ samples independently from $q$. The defender, upon observing the $n$-length word generated by the external agent, wants to detect the presence of the attacker.

Throughout the paper, a decision rule implemented by the defender is denoted by $\varphi : \mathcal{X}^n \to [0, 1]$, with the interpretation that $\varphi(\mathbf{x}^n)$ is the probability with which hypothesis $H_1$ is accepted (i.e., the presence of an adversary is declared) when the defender observes the $n$-length word $\mathbf{x}^n = (x_1, \ldots, x_n)$. We say that a decision rule $\varphi$ is deterministic if $\varphi(\mathbf{x}^n) \in \{0, 1\}$ for all $\mathbf{x}^n \in \mathcal{X}^n$.

To define the game, let the attacker's strategy set be $Q \subseteq M_1(\mathcal{X})$, and that of the defender be

$$\Phi_n = \{\varphi : \mathcal{X}^n \to [0, 1]\},$$

which is the set of all randomized decision rules on $n$-length words.

To define the utilities, consider the attacker first. We assume that there is a cost associated with choosing a distribution from $Q$ which we model using a cost function $c : Q \to \mathbb{R}_+$. The goal of the attacker is to fool the defender as much as possible, i.e., he wants to maximize the probability that the defender classifies an $n$-length word as coming from the non-attacker whereas it is actually being generated by the attacker. To capture this, the utility of the attacker when she plays the pure strategy $q \in Q$ and the defender plays the pure strategy $\varphi \in \Phi_n$ is defined as

$$u_n^A(q, \varphi) = \sum_{\mathbf{x}^n} (1 - \varphi(\mathbf{x}^n)) q(\mathbf{x}^n) - c(q), \tag{3.1}$$

where $q(\mathbf{x}^n)$ denotes the probability of observing the $n$-length word $\mathbf{x}^n$ when the symbols are generated independently from the distribution $q$.

For the defender, the goal is to minimize the classification error. Similar to the classical hypothesis testing problem, there could be two types of errors: (i) the external agent is actually a non-attacker whereas the defender declares that there is an attack (Type I error, or false alarm), and (ii) the external agent is an attacker whereas the defender declares that there is no attack (Type II error, or missed detection). The goal of the defender is to minimize a weighted sum of the above two types of errors. After suitable normalization, we define the utility of the defender as

$$u_n^D(q, \varphi) = -\left( \sum_{\mathbf{x}^n} (1 - \varphi(\mathbf{x}^n)) q(\mathbf{x}^n) + \gamma \sum_{\mathbf{x}^n} \varphi(\mathbf{x}^n) p(\mathbf{x}^n) \right), \tag{3.2}$$

where $\gamma > 0$ is a constant that captures the exogenous probability of attack (i.e., $\theta$), as well as the relative weights given to the error terms.

We denote our Bayesian hypothesis testing game with utility functions (3.1) and (3.2) by $\mathcal{G}^B(d, n)$. With a slight abuse of notation, we denote by $u_n^A(\sigma_n^A, \sigma_n^D)$ and $u_n^D(\sigma_n^A, \sigma_n^D)$, the utility of the players under a mixed strategy $(\sigma_n^A, \sigma_n^D)$, where $\sigma_n^A \in M_1(Q)$, and $\sigma_n^D \in M_1(\Phi_n)$.

For our analysis of game $\mathcal{G}^B(d, n)$, we will make use of the following assumptions:

(A1)  $Q$ is a closed subset of $M_1(\mathcal{X})$, and $p \notin Q$.

(A2)  $p(i) > 0$ for all $i \in \mathcal{X}$. Furthermore, for each $q \in Q$, $q(i) > 0$ for all $i \in \mathcal{X}$.

(A3)  $c$ is continuous on $Q$, and there exists a unique $q^* \in Q$ such that

$$q^* = \arg\min_{q \in Q} c(q).$$

(A4)  The point $p$ is distant from the set $Q$ relative to the point $q^*$, i.e.,

$$\{\mu \in M_1(\mathcal{X}) : D(\mu||p) \le D(\mu||q^*)\} \cap Q = \emptyset,$$

where $D(\mu||\nu) = \sum_{i \in \mathcal{X}} \mu(i) \log \frac{\mu(i)}{\nu(i)}, \mu, \nu \in M_1(\mathcal{X})$, denotes the relative entropy between the distributions $\mu$ and $\nu$.

Note that (A1) and (A2) are very natural. In (A2), if $p(i) = 0$ for some $i \in \mathcal{X}$ and $q(i) > 0$ for some $q \in Q$, then the adversary will never pick $q$, as the defender can easily detect the presence of the attacker by looking for element $i$. On the other hand, if $p(i) = 0$ and $q(i) = 0$ for all $q \in Q$, we may consider a new alphabet set without $i$. In (A3), continuity of the cost function $c$ is natural and we do not assume any extra condition other than the requirement that there is a unique minimizer. Assumption (A4) is used to show certain property of the equilibrium of the defender, which is later used in the study of error exponents associated with classification error. Specifically, Assumption (A4) is used in the proofs of Lemma 4.4, Lemma 4.5 and Theorem 4.1; all other results are valid without this assumption. We will further discuss the role of (A3) and (A4) in Section 4.3 after Theorem 4.1.

## 3.2  Model discussion

Our setting is that of adversarial hypothesis testing, where the attacker chooses a distribution and points are then generated i.i.d. according to it. This is a reasonable model in applications such as multimedia forensics (where one tries to determine if an attacker has tampered with an image from signals that can be modeled as random variables following an image-dependent distribution) or biometrics (where again one tries to detect from random signals whether the perceived signals do come from the characteristics of a given individual or they come from tampered characteristics)—see more details about these applications in [2, 3, 26]. In such applications, it is reasonable that different ways of tampering have different costs for the attacker and that one can estimate those costs for a given application at least to some extent. Modeling the attacker's utility via a cost function is classical in other settings, for instance in adversarial classification [12, 25, 6] and experiments with real-world applications where a reasonable cost function can be estimated has been done, for instance, in [6].

Our setting is very similar to that of a composite hypothesis testing framework where nature picks a distribution from a given set and generates independent samples from it. However, in such problems, one does not model a utility function for the nature/statistician and one is often interested in existence and properties of uniformly most powerful test or locally most powerful test (depending on the Bayesian or frequentist approach; see Section II.E in [23]). In contrast, here, we specifically model the utility functions for the agents and investigate the behavior at Nash equilibrium using very different analysis, which is more natural in adversarial settings where two rational agents interact.

Our setting also coincides with the well-studied setting of minimax testing [15] when $c(q) = 0$ for all $q \in Q$ (and hence every $q$ is a minimizer of $c$). Note, however, that this case is not included in our model due to Assumption (A3)—rather we study the opposite extreme where $c$ has a unique minimizer. Our results are not an easy extension of the classical results because our game is now a nonzero-sum game (whereas the minimax setting corresponds to a zero-sum game). We can therefore not inherit any of the nice properties of zero-sum games; in particular we cannot compute the NE and we instead have to prove properties of the NE (e.g., concentration) without being able to explicitly compute it. In fact, our results too are quite different since we show that the error rate is the same as a simple test where $H_1$ would contain only $q^*$, which is different from the classical minimax case.

Finally, in our model we fix the sample size $n$, i.e., the defender makes a decision only after observing all $n$ samples. We restrict to this simpler setting since it has applications in various domains (see Section 1.1), and understanding the equilibrium of such games leads to interesting and non-trivial results. We leave the study of a sequential model where the defender has the flexibility to choose the number of samples for decision making as future work.

## 4 Main Results

### 4.1 Mixed Strategy Nash Equilibrium for $\mathcal{G}^B(d, n)$

We first examine the Nash equilibrium for $\mathcal{G}^B(d, n)$. Note that the strategy sets of both the attacker and the defender are uncountable, hence it is a priori not clear whether our game has a Nash equilibrium.

Towards this, we equip the set $\Phi_n$ of all randomized decision rules with the sup-norm metric, i.e.,

$$d_n(\varphi_1, \varphi_2) = \max_{\mathbf{x}^n \in \mathcal{X}^n} |\varphi_1(\mathbf{x}^n) - \varphi_2(\mathbf{x}^n)|,$$

for $\varphi_1, \varphi_2 \in \Phi_n$. It is easy to see that the set $\Phi_n$ endowed with the above metric is a compact metric space. We also equip $M_1(\mathcal{X})$ with the usual Euclidean topology on $\mathbb{R}^d$, and equip $Q$ with the subspace topology. Also, for studying the mixed extension of the game, we equip the spaces $M_1(Q)$ and $M_1(\Phi_n)$ with their corresponding weak topologies. Product spaces are always equipped with the corresponding product topology.

We begin with a simple continuity property of the utility functions.

**Lemma 4.1.** *Assume (A1)-(A3). Then, the utility functions $u_n^A$ and $u_n^D$ are continuous on $Q \times \Phi_n$.*

We now show the main result of this subsection, namely existence and partial characterization of a NE for our hypothesis testing game.

**Proposition 4.1.** *Assume (A1)-(A3). Then, there exists a mixed strategy Nash equilibrium for $\mathcal{G}^B(d, n)$. If $(\hat{\sigma}_n^A, \hat{\sigma}_n^D)$ is a NE, then so is $(\hat{\sigma}_n^A, \hat{\varphi}_n)$ where $\hat{\varphi}_n$ is the likelihood ratio test given by*

$$\hat{\varphi}_n(\mathbf{x}^n) = \begin{cases} 1, & \text{if } q_{\hat{\sigma}_n^A}(\mathbf{x}^n) - \gamma p(\mathbf{x}^n) > 0, \\ \varphi_{\hat{\sigma}_n^D}, & \text{if } q_{\hat{\sigma}_n^A}(\mathbf{x}^n) - \gamma p(\mathbf{x}^n) = 0, \\ 0, & \text{if } q_{\hat{\sigma}_n^A}(\mathbf{x}^n) - \gamma p(\mathbf{x}^n) < 0, \end{cases} \tag{4.1}$$

*where $q_{\hat{\sigma}_n^A}(\mathbf{x}^n) = \int q(\mathbf{x}^n)\hat{\sigma}_n^A(dq)$, and $\varphi_{\hat{\sigma}_n^D} = \int \varphi(\mathbf{x}^n)\hat{\sigma}_n^D(d\varphi)$.*

The existence of a NE follows from Glicksberg's fixed point theorem (see e.g., Corollary 2.4 in [24]); for the form of the defender's equilibrium strategy, we have to examine the utility function $u_n^D$.

*Remark* 4.1. Note that we have considered randomization over $\Phi_n$ to show existence of a NE. Once this is established, we can then show the form of the strategy of the defender $\hat{\varphi}_n$ at equilibrium; the existence of a NE is not clear if we do not consider randomization over $\Phi_n$.

*Remark* 4.2. Note that the distribution $q_{\hat{\sigma}_n^A}$ on $\mathcal{X}^n$ cannot necessarily be written as an $n$-fold product distribution of some element from $M_1(\mathcal{X})$. Therefore, the test $\hat{\varphi}_n$ is slightly different from the usual likelihood ratio test that appears in the classical hypothesis testing problem where samples are generated independently.

*Remark* 4.3. Apart from the conditions of the above proposition, a sufficient condition for existence of pure strategy Nash equilibrium is that the utilities are individually quasiconcave, i.e., $u_n^A(\cdot, \varphi)$ is quasiconcave for all $\varphi \in \Phi_n$, and $u_n^D(q, \cdot)$ is quasiconcave for all $q \in Q$. However, it is easy to check that the Type II error term is not quasiconcave in the attacker's strategy, and hence the utility of the attacker is not quasiconcave. Hence, a pure strategy Nash equilibrium is not guaranteed to exist—see numerical experiments in Appendix C of the full version of this paper [27].

*Remark* 4.4. Proposition 4.1 does not provide any information about the structure of the attacker's strategy at a NE. We believe that obtaining the complete structure of a NE and computing it is a difficult problem in general because the strategy spaces of both players are uncountable (and there is no pure-strategy NE in general), and we cannot use the standard techniques for finite games. However, we emphasize that we are able to obtain error exponents at an equilibrium (see Theorem 4.1) without explicitly computing the structure of a NE. Also, one could study Stackelberg equilibrium for our game $\mathcal{G}^B(d, n)$ to help solve computational issues, although we note that most of the security games literature using Stackelberg games assumes finite action spaces (see, for example, [17]); however we do not address the study of Stackelberg equilibrium in this paper.

### 4.2 Concentration Properties of Equilibrium

We now study some concentration properties of the mixed strategy Nash equilibrium for the game $\mathcal{G}^B(d, n)$ for large $n$. The results in this section will be used later to show the exponential convergence of the classification error at equilibrium.

Let $e_n$ denote the classification error, i.e., $e_n(q, \varphi) = -u_n^D(q, \varphi), q \in Q, \varphi \in \Phi_n$. We begin with the following lemma, which asserts that the error at equilibrium is small for large enough $n$.

**Lemma 4.2.** *Assume (A1)-(A3). Let $(\hat{\sigma}_n^A, \hat{\sigma}_n^D)_{n \geq 1}$ be a sequence such that, for each $n \geq 1$, $(\hat{\sigma}_n^A, \hat{\sigma}_n^D)$ is a mixed strategy Nash equilibrium for $\mathcal{G}^B(d, n)$. Then, $e_n(\hat{\sigma}_n^A, \hat{\sigma}_n^D) \to 0$ as $n \to \infty$.*

The main idea in the proof is to let the defender play a decision rule whose acceptance set is a small neighborhood around the point $p$, and then bound $e_n(\hat{\sigma}_n^A, \hat{\sigma}_n^D)$ using the error of the above strategy.

We now show that the mixed strategy profile of the attacker $\hat{\sigma}_n^A$ converges weakly to the point mass at $q^*$ (denoted by $\delta_{q^*}$) as $n \to \infty$. This is a consequence of the fact that $q^*$ is the minimizer of $c$, and hence for large enough $n$, the attacker does not gain much by deviating from the point $q^*$.

**Lemma 4.3.** *Assume (A1)-(A3), and let $(\hat{\sigma}_n^A, \hat{\sigma}_n^D)_{n \geq 1}$ be as in Lemma 4.2. Then, $\hat{\sigma}_n^A \to \delta_{q^*}$ weakly as $n \to \infty$.*

Note that it is not clear from the above lemma that the equilibrium strategy of the attacker $\hat{\sigma}_n^A$ is supported on a small neighborhood around $q^*$ for large enough $n$. By playing a strategy $q$ that is far from $q^*$ we could still have $u_n^A(q, \hat{\sigma}_n^D) = u_n^A(\hat{\sigma}_n^A, \hat{\sigma}_n^D)$, since the error term in $u_n^A$ could compensate for the possible loss of utility from the cost term. We now proceed to show that this cannot happen under Assumption (A4). We first argue that the equilibrium error is small even when the attacker deviates from her equilibrium strategy.

**Lemma 4.4.** *Assume (A1)-(A4), and let $(\hat{\sigma}_n^A, \hat{\sigma}_n^D)_{n \geq 1}$ be as in Lemma 4.2. Then,*

$$\sup_{q \in Q} e_n(q, \hat{\sigma}_n^D) \to 0 \text{ as } n \to \infty.$$

We are now ready to show the concentration of the attacker's equilibrium:

**Lemma 4.5.** *Assume (A1)-(A4), and let $(\hat{\sigma}_n^A, \hat{\sigma}_n^D)_{n \geq 1}$ be as in Lemma 4.2. Let $(q_n)_{n \geq 1}$ be a sequence such that $q_n \in supp(\hat{\sigma}_n^A)$ for each $n \geq 1$. Then, $q_n \to q^*$ as $n \to \infty$.*

The above concentration phenomenon is a consequence of the uniqueness of $q^*$ and Assumption (A4). The main idea in the proofs of Lemma 4.4 and Lemma 4.5 is to essentially show that, for large enough $n$, the acceptance region of $H_0$ under (any) mixed strategy Nash equilibrium does not intersect the set $Q$. If we do not assume (A4), then the decision region at equilibrium could intersect $Q$, and we may not have the concentration property in the above lemma (we will still have the convergence property in Lemma 4.3 though, which does not use (A4)).

### 4.3 Error Exponents

With the results on concentration properties of the equilibrium from the previous section, we are now ready to examine the error exponent associated with the classification error at equilibrium. Let $\Lambda_0$ denote the log-moment generating function of the random variable $\frac{q^*(X)}{p(X)}$ under $H_0$, i.e., when $X \sim p$: $\Lambda_0(\lambda) = \log \sum_{i \in \mathcal{X}} \exp\left(\lambda \frac{q^*(i)}{p(i)}\right) p(i)$, $\lambda \in \mathbb{R}$. Define its Fenchel-Legendre transform

$$\Lambda_0^*(x) = \sup_{\lambda \in \mathbb{R}} \{\lambda x - \Lambda_0(\lambda)\}, \ x \in \mathbb{R}.$$

Our main result in the paper (for the Bayesian case) is the following theorem.

**Theorem 4.1.** *Assume (A1)-(A4), and let $(\hat{\sigma}_n^A, \hat{\sigma}_n^D)_{n \geq 1}$ be as in Lemma 4.2. Then,*

$$\lim_{n \to \infty} \frac{1}{n} \log e_n(\hat{\sigma}_n^A, \hat{\sigma}_n^D) = -\Lambda_0^*(0).$$

Our approach to show this result is via obtaining asymptotic lower and upper bounds for the classification error at equilibrium $e_n(\hat{\sigma}_n^A, \hat{\sigma}_n^D)$. Since we do not have much information about the structure of the equilibrium, we first let one of the players deviate from their equilibrium strategy, so that we can estimate the error corresponding to the new pair of strategies, and then use these estimates to compute the error rate at equilibrium. The lower bound easily follows by letting the attacker play the strategy $q^*$, and using the error exponent in the classical hypothesis testing problem between $p$ and $q^*$. For the upper bound, we let the defender play a specific deterministic decision rule, and make use of the concentration properties of the equilibrium of the attacker in Section 4.2.

Thus, we see that the error exponent is the same as that for the classical hypothesis testing problem of $X_1, \ldots, X_n$ i.i.d.$\sim p$ versus $X_1, \ldots, X_n$ i.i.d.$\sim q^*$ (see Corollary 3.4.6 in [10]). That is, for large values of $n$, the adversarial hypothesis testing game is not much different from the above classical setting (whose parameters are derived from the adversarial setting) in terms of classification error. We emphasize that we have not used any property of the specific structure of the mixed strategy Nash equilibrium in obtaining the error exponent associated with the classification error, and hence Theorem 4.1 is valid for any NE. We believe that obtaining the actual structure of a NE is a difficult problem, as the strategy spaces are infinite, and the utility functions do not possess any monotonicity properties in general. For numerical computation of error exponents in a simple case, see Section 5.

We conclude this section by discussing the role of Assumptions (A3) and (A4). We used (A4) to obtain the concentration of equilibrium in Lemma 4.5. Without this assumption, Theorem 4.1 is not valid; see Section 5 for numerical counter-examples. Also, in our setting, unlike the classical minimax testing, it is not clear whether it is always true that the error goes to $0$ as the number of samples becomes large, and whether the attacker should always play a point close to $q^*$ at equilibrium. It could be that playing a point far from $q^*$ is better if she can compensate the loss from $c$ from the error term. In fact, that is what happens when (A4) is not satisfied, since there is partial overlap of the decision region of the defender with the set $Q$. Regarding (A3), when $c$ has multiple minimizers, our analysis can only tell us that the equilibrium of the attacker is supported around the set of minimizers for large enough $n$; to study error exponents in such cases, one has to do a finer analysis of characterizing the attacker's equilibrium. All in all, using (A3) and (A4) allows us to establish interesting concentration properties of the equilibrium (which is not a priori clear) and error exponents associated with classification error without characterizing a NE, hence we believe that these assumptions serve as a good starting point.

## 5   Numerical Experiments

In this section, we present two numerical examples in the Bayesian formulation to illustrate the result in Theorem 4.1 and the importance of Assumption (A4). Due to space limitations, additional experiments in the Bayesian formulation are relegated to Appendix C of the full version [27], which illustrate (a) best response strategies of the players, (b) existence of pure strategy Nash equilibrium for large values on $n$ as suggested by Lemma 4.5, and (c) importance of Assumption (A4).[1]

We illustrate the result in Theorem 4.1 numerically in the following setting. We fix $\mathcal{X} = \{0, 1\}$ (i.e. $d = 2$) and each probability distribution on $\mathcal{X}$ is represented by the probability that it assigns to the symbol 1, and hence $M_1(\mathcal{X})$ is viewed as the unit interval. We fix $p = 0.5$. For numerical computations, we discretize the set $Q$ into 100 equally spaced points, and we only consider deterministic threshold-based decision rules for the defender. To compute a NE, we solve the linear programs associated with attacker as well as the defender for the zero-sum equivalent game of $\mathcal{G}^B(2, n)$.

For the function $c(q) = |q - q^*|$ with $q^* = 0.8$, Figure 1(a) shows the error exponent at the NE computed by the above procedure as a function of the number of samples, from $n = 10$ to $n = 300$ in steps of 10. As suggested by Theorem 4.1, we see that the error exponents approach the value $\Lambda_0^*(0) = 0.054$ (the boundary of the decision region is around the point $q = 0.66$, and $D(q||p) \approx D(q||q^*) \approx 0.054$).

We now consider an example which demonstrates that, the result on error exponent in Theorem 4.1 may not be valid if Assumption (A4) is not satisfied. In this experiment, we consider the case where $Q = [0.6, 0.9]$ and $q^* = 0.9$. Note that the present setting does not satisfy Assumption (A4). Figure 1(b) shows the error exponent at the equilibrium as a function of $n$, from $n = 100$ to $n = 400$ in steps of 100, for the cost function $c(q) = 3|q - q^*|$. From this plot, we see that, the error exponents converge to somewhere around 0.032, whereas $\Lambda_0^*(0) \approx 0.111$.

## 6   Concluding Remarks

In this paper, we studied hypothesis testing games that arise in the context of adversarial classification. We showed that, at equilibrium, the strategy of the classifier is to use a likelihood ratio test. We also

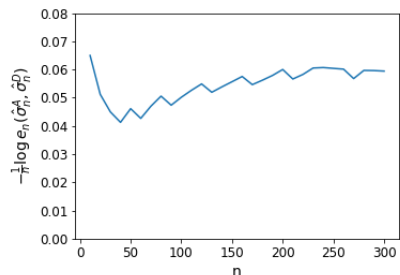
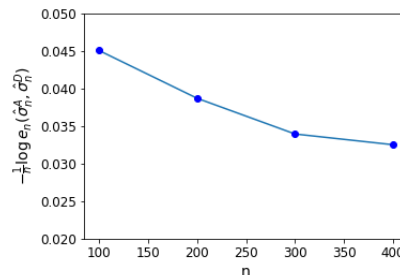

(a) $Q = [0.7, 0.9], c(q) = |q - 0.8|$    (b) $Q = [0.6, 0.9], c(q) = 3|q - 0.9|$

Figure 1: Error exponents as a function of $n$

examined the exponential rate of decay of classification error at equilibrium and showed that it is same as that of a classical testing problem with parameters derived from the adversarial model.

Throughout the paper, we assumed that the alphabet $\mathcal{X}$ is finite. This is a reasonable assumption in applications that deal with digital signals such as image forensics (an important application for adversarial hypothesis testing); and it is also a good starting point because even in this case, our analysis of the error exponents is nontrivial. Making $\mathcal{X}$ countable/uncountable will make the space $M_1(\mathcal{X})$ infinite dimensional, and the analysis of error exponents will become more difficult (e.g., the continuity of relative entropy is no longer true in this case, which we crucially use in our analysis), but the case of general state space $\mathcal{X}$ is an interesting future direction.

Finding the exact structure of the equilibrium for our hypothesis testing games is a challenging future direction. This will also shed some light on the error exponent analysis for the case when Assumption (A4) is not satisfied. Another interesting future direction is to examine the hypothesis testing game in the sequential detection context where the defender can also decide the number of data samples for classification. In such a setting, an important question is to understand whether the optimal strategy of the classifier is to use a standard sequential probability ratio test.

### Acknowledgments

The first author is partially supported by the Cisco-IISc Research Fellowship grant. The work of the second author was supported in part by the French National Research Agency (ANR) through the "Investissements d'avenir" program (ANR-15-IDEX-02) and through grant ANR-16- TERC0012; and by the Alexander von Humboldt Foundation.

## Footnotes

[1]Appendix C of the full version [27] also contains numerical experiments in the Neyman-Pearson formulation presented in Appendix A of the full version [27]. The code used for our simulations is available at `https://github.com/sarath1789/ahtg_neurips2019`.

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
