[Supplementary Material]

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

# A   Hypothesis Testing Game: Neyman-Pearson Formulation

In this section, we study the Neyman-Pearson version of the hypothesis testing problem. The presentation of results in this section is similar to the Bayesian case, and we will also use the same notation as in the Bayesian formulation.

## A.1   Problem Formulation

In the Neyman-Pearson point of view, we do not assume any knowledge on the probability that the external agent is an attacker. Fix $\varepsilon > 0$. As before, the strategy set of the attacker is the set $Q$. For the defender, motivated by the Neyman-Pearson approach for the classical hypothesis testing problem, we define the strategy set as the set of all randomized decision rules on $n$-length words whose false alarm probability is at most $\varepsilon$, i.e.,

$$\Phi_n = \{\varphi : \mathcal{X}^n \to [0,1] : P_n^{FA}(\varphi) \leq \varepsilon\},$$

where $P_n^{FA}(\varphi) = \sum_{\mathbf{x}^n} \varphi(\mathbf{x}^n)p(\mathbf{x}^n)$ denotes the false alarm probability under the decision rule $\varphi$.

We now define the utilities. Similar to the Bayesian framework, the utility of the attacker is defined as

$$u_n^A(q, \varphi) = \sum_{\mathbf{x}^n}(1 - \varphi(\mathbf{x}^n))q(\mathbf{x}^n) - c(q). \tag{A.1}$$

Since we have already constrained the strategy set of the defender by imposing an upper bound on the Type I error, the utility of the defender is defined as

$$u_n^D(q, \varphi) = -\left(\sum_{\mathbf{x}^n}(1 - \varphi(\mathbf{x}^n))q(\mathbf{x}^n)\right), \tag{A.2}$$

which captures the Type II error.

We denote our two-player Neyman-Pearson hypothesis testing game with utility functions (A.1) and (A.2) by $\mathcal{G}^{NP}(\varepsilon, d, n)$. Similar to the Bayesian case, we will use Assumptions (A1)-(A4) throughout the analysis of the Neyman-Pearson case.

## A.2   Mixed Strategy Nash Equilibrium for $\mathcal{G}^{NP}(\varepsilon, d, n)$

We now examine the mixed strategy Nash equilibrium for the game $\mathcal{G}^{NP}(\varepsilon, d, n)$. Similar to the Bayesian framework, we endow the set $Q$ with the standard Euclidean topology on $\mathbb{R}^d$ and the set $\Phi_n$ with the sup-norm metric. The following lemma asserts compactness of the strategy space of the defender.

**Lemma A.1.** *The set $\Phi_n$ equipped with the metric $d_n$ is a compact metric space.*

We also have joint continuity of the utility functions of both the players, and it can be proved similar to Lemma 4.1.

**Lemma A.2.** *Assume (A1)-(A3). Then, the utility functions $u_n^A$ and $u_d^D$ are continuous on $Q \times \Phi_n$.*

We are now ready to show the existence of a mixed strategy Nash equilibrium.

**Proposition A.1.** *Assume (A1)-(A3). Then, there exists a mixed strategy Nash equilibrium for $\mathcal{G}^{NP}(\varepsilon, d, n)$. If $(\hat{\sigma}_n^A, \hat{\sigma}_n^D)$ is a NE, then $\hat{\sigma}_n^D$ is the point mass at the most powerful $\varepsilon$-level Neyman-Pearson test for $X_1, \ldots, X_n$ i.i.d. $\sim p$ versus $(X_1, \ldots, X_n) \sim q_{\hat{\sigma}_n^A}$.*

The existence of a mixed strategy equilibrium easily follows from the compactness of strategy spaces and continuity of utility functions. To show the specific form of defender's equilibrium strategy, we appeal to the Neyman-Pearson lemma.

Let $(\hat{\sigma}_n^A, \hat{\varphi}_n)$ denote a NE given by Proposition A.1.

## A.3 Characterization of Equilibrium in the Binary case

Consider the game $\mathcal{G}^{NP}(\varepsilon, d, n)$ in the binary case, i.e., $d = 2$. Here, there are some interesting monotonicity properties of the utility functions that allow us to get a pure strategy Nash equilibrium for $\mathcal{G}^{NP}(\varepsilon, 2, n)$, in which the defender plays a threshold-based test, i.e., declares the presence of an adversary whenever the number of 1's in the observation exceeds a threshold:

**Lemma A.3.** *Assume (A1)-(A3). Then, the defender admits a strictly dominant strategy, and there exists a pure strategy Nash equilibrium for $\mathcal{G}^{NP}(\varepsilon, 2, n)$.*

*Remark* A.1. The monotonicity alluded to above is a consequence of the fact that $u_n^D$ captures just the Type II error. In the Bayesian framework, we do not have this monotonicity in $u_n^D$, due to the presence of both the Type I and Type II errors in $u_n^D$, and hence, existence of a pure strategy Nash equilibrium in the binary case cannot be guaranteed in the Bayesian framework.

## A.4 Concentration Properties of Equilibrium

In this section, we study some concentration properties of the equilibrium. We have the following two lemmas, which can be proved similar to the corresponding Lemmas for the Bayesian formulation in Section 4.2.

**Lemma A.4.** *Assume (A1)-(A3). Let $(\hat{\sigma}_n^A, \hat{\varphi}_n)_{n \geq 1}$ be a sequence such that, for each $n \geq 1$, $(\hat{\sigma}_n^A, \hat{\varphi}_n)$ is a mixed strategy Nash equilibrium for $\mathcal{G}^{NP}(\varepsilon, d, n)$. Then, $e_n(\hat{\sigma}_n^A, \hat{\varphi}_n) \to 0$ as $n \to \infty$.*

**Lemma A.5.** *Assume (A1)-(A3), and let $(\hat{\sigma}_n^A, \hat{\varphi}_n)_{n \geq 1}$ be as in Lemma A.4. Then $\hat{\sigma}_n^A \to \delta_{q^*}$ weakly as $n \to \infty$.*

We also have that the error at equilibrium goes to 0 even when the attacker deviates from her equilibrium strategy.

**Lemma A.6.** *Assume (A1)-(A4), and let $(\hat{\sigma}_n^A, \hat{\varphi}_n)_{n \geq 1}$ be as in Lemma A.4. Then,*

$$\sup_{q \in Q} e_n(q, \hat{\varphi}_n) \to 0 \text{ as } n \to \infty.$$

The main idea in the proof of the above lemma is to show that the acceptance region of $H_0$ under any equilibrium does not intersect the set $Q$. With this lemma at hand, we now have the following concentration property for the support of the equilibrium strategy of the attacker, which can be proved similar to Lemma 4.5 in the Bayesian formulation.

**Lemma A.7.** *Assume (A1)-(A4), and let $(\hat{\sigma}_n^A, \hat{\varphi}_n)_{n \geq 1}$ be as in Lemma A.4. Let $(q_n)_{n \geq 1}$ be a sequence such that $q_n \in supp(\hat{\sigma}_n^A)$ for each $n \geq 1$. Then, $q_n \to q^*$ as $n \to \infty$.*

*Remark* A.2. Note that, in one dimension ($d = 1$), the acceptance region of an optimal Neyman-Pearson test for a fixed alternative $q$ will be a "vanishingly small neighborhood of the null distribution $p$" and that while it can still intersect $Q$ for finite $n$, it may not for large-enough $n$; so that Lemma A.6 may always hold. However, it is unclear how this might generalize to higher dimension. Intuitively, in higher dimension, the acceptance region may become close to $p$ only from certain directions. We also note that our proof of Lemma A.6 actually uses Assumption (A4) and not a weaker version of it—see the expression of $\Gamma_n$ in the proof of Lemma A.6. Therefore, we believe that (A4) is needed in higher dimensions even for the Neyman-Pearson case; although it is possible that a weaker assumption will suffice in one dimension.

## A.5 Error Exponents

Our main result in the Neyman-Pearson formulation is the following theorem.

**Theorem A.1.** *Assume (A1)-(A4), and let $(\hat{\sigma}_n^A, \hat{\varphi}_n)_{n \geq 1}$ be as in Lemma A.4. Then,*

$$\lim_{n \to \infty} \frac{1}{n} \log e_n(\hat{\sigma}_n^A, \hat{\varphi}_n) = -D(p||q^*).$$

Again, we note that the error exponent is the same as that of the classical Neyman-Pearson hypothesis testing problem between $p$ and $q^*$.

# B Proofs

## B.1 Proof of Lemma 4.1

Since we are on a metric space, it suffices to show sequential continuity. Let $\{(q_k, \varphi_k), k \geq 1\}$ be a sequence such that $(q_k, \varphi_k) \to (q, \varphi)$ as $k \to \infty$. First, consider $u_n^D$. Notice that, for each $\mathbf{x}^n$, we have $q_k(\mathbf{x}^n) \to q(\mathbf{x}^n)$, and $\varphi_k(\mathbf{x}^n) \to \varphi(\mathbf{x}^n)$ as $k \to \infty$. Therefore, we have that $q_k(\mathbf{x}^n)\varphi_k(\mathbf{x}^n) \to q(\mathbf{x}^n)\varphi(\mathbf{x}^n)$, which yields that,

$$\lim_{k \to \infty} \sum_{\mathbf{x}^n} q_k(\mathbf{x}^n)\varphi_k(\mathbf{x}^n) = \sum_{\mathbf{x}^n} q(\mathbf{x}^n)\varphi(\mathbf{x}^n).$$

Similarly, we also have

$$\lim_{k\to\infty} \sum_{\mathbf{x}^n} p(\mathbf{x}^n)\varphi_k(\mathbf{x}^n) = \sum_{\mathbf{x}^n} p(\mathbf{x}^n)\varphi(\mathbf{x}^n).$$

Therefore, we have that $u_n^D(q_k, \varphi_k) \to u_n^D(q, \varphi)$ as $k \to \infty$ which proves continuity of the utility of the defender. Using similar arguments, and by using the continuity of the cost function $c$ on $Q$, we see that $u_n^A(q_k, \varphi_k) \to u_n^A(q, \varphi)$ as $k \to \infty$, which shows the continuity of the utility of the attacker. $\qquad\square$

## B.2 Proof of Proposition 4.1

$\mathcal{G}^B(d, n)$ is a two-player game with compact strategy spaces. Also, by Lemma 4.1, the utilities (in pure strategies) of both the attacker and the defender are jointly continuous on $Q \times \Phi_n$. Therefore, an application of the Glicksberg fixed point theorem (see, for example, Corollary 2.4 in [26]) tell us that there exists a mixed strategy Nash equilibrium (NE) for the adversarial hypothesis testing game $\mathcal{G}^B(d, n)$.

We now show the structure of the equilibrium strategy of the defender. Note that, for any $\varphi \in \Phi_n$, $u_n^D(\hat{\sigma}_n^A, \varphi) = -1 + \sum_{\mathbf{x}^n} \varphi(\mathbf{x}^n)(q_{\hat{\sigma}_n^A}(\mathbf{x}^n) - \gamma p(\mathbf{x}^n))$, where $q_{\hat{\sigma}_n^A}(\mathbf{x}^n) = \int q(\mathbf{x}^n)\hat{\sigma}_n^A(dq)$. Therefore, using the characterization of a NE (see Proposition 140.1 in [23]), it follows that for any $\varphi \in \text{supp}(\hat{\sigma}_n^D)$, we have

$$\varphi(\mathbf{x}^n) = \begin{cases} 1, & \text{if } q_{\hat{\sigma}_n^A}(\mathbf{x}^n) - \gamma p(\mathbf{x}^n) > 0, \\ 0, & \text{if } q_{\hat{\sigma}_n^A}(\mathbf{x}^n) - \gamma p(\mathbf{x}^n) < 0. \end{cases}$$

Now, define $\hat{\varphi}_n$ such that $\hat{\varphi}_n(\mathbf{x}^n) = \int \varphi(\mathbf{x}^n)\hat{\sigma}_n^D(d\varphi)$ whenever $\mathbf{x}^n$ is such that $q_{\hat{\sigma}_n^A}(\mathbf{x}^n) - \gamma p(\mathbf{x}^n) = 0$, and that satisfies the above condition when $\mathbf{x}^n$ is such that $q_{\hat{\sigma}_n^A}(\mathbf{x}^n) - \gamma p(\mathbf{x}^n) \neq 0$. Consider the strategy profile $(\hat{\sigma}_n^A, \hat{\varphi}_n)$ where the defender plays the pure strategy $\hat{\varphi}_n$. By the choice of $\hat{\varphi}_n$, we see that $u_n^A(q, \hat{\varphi}_n) = u_n^A(q, \hat{\sigma}_n^D)$ for all $q \in Q$, and $u_n^D(\hat{\sigma}_n^A, \varphi) \leq u_n^D(\hat{\sigma}_n^A, \hat{\varphi}_n)$ for any $\varphi \in \Phi_n$. Therefore, using the characterization of a NE, we see that $(\hat{\sigma}_n^A, \hat{\varphi}_n)$ is a NE. This completes the proof of the Proposition. $\qquad\square$

## B.3 Proof of Lemma 4.2

By Assumption (A1), there exist a $\delta > 0$ such that $B(p, \delta) \cap Q = \emptyset$, where $B(p, \delta)$ denotes an open ball of radius $\delta$ centered at $p$. Let $\varphi^\delta$ denote the deterministic decision rule whose rejection region is the set $B(p, \delta)^c$, i.e., $\varphi^\delta(\mathbf{x}^n) = 1$ whenever $\mathcal{P}_{\mathbf{x}^n} \in B(p, \delta)^c$ and $\varphi^\delta(\mathbf{x}^n) = 0$ otherwise, where $\mathcal{P}_{\mathbf{x}^n} \in M_1(\mathcal{X})$ denotes the type of $\mathbf{x}^n$, i.e., $\mathcal{P}_{\mathbf{x}^n}(i) = \frac{1}{n}\sum_{k=1}^n 1_{\{x_k=i\}}$. Since $(\hat{\sigma}_n^A, \hat{\sigma}_n^D)$ is a Nash equilibrium, and $e_n(\hat{\sigma}_n^A, \hat{\sigma}_n^D) = -u_n^D(\hat{\sigma}_n^A, \hat{\sigma}_n^D)$, we see that

$$e_n(\hat{\sigma}_n^A, \hat{\sigma}_n^D) \leq e_n(\hat{\sigma}_n^A, \varphi^\delta), \tag{B.1}$$

where $(\hat{\sigma}_n^A, \varphi^\delta)$ denotes the strategy profile where the attacker plays the mixed strategy $\hat{\sigma}_n^A$ and the defender plays the pure strategy $\varphi^\delta$.

We now proceed to bound the error term $e_n(\hat{\sigma}_n^A, \varphi^\delta)$. We have

$$e_n(\hat{\sigma}_n^A, \varphi^\delta) = \int \left[ \sum_{\mathbf{x}^n} (1 - \varphi^\delta(\mathbf{x}^n))q(\mathbf{x}^n) + \gamma\varphi^\delta(\mathbf{x}^n)p(\mathbf{x}^n) \right] \hat{\sigma}_n^A(dq)$$

$$= \int q(\mathcal{P}_{\mathbf{x}^n} \in B(p, \delta))\hat{\sigma}_n^A(dq) + \gamma p(\mathcal{P}_{\mathbf{x}^n} \in B(p, \delta)^c).$$

We bound the first term above using a simple upper bound for the probability of observing a given type under a given distribution (see, for example, Lemma 2.1.9 in [10]). Let $\mathcal{P}_n$ denote the set of all possible types of an $n$-length word. For any $q \in Q$, we have that

$$q(\mathcal{P}_{\mathbf{x}^n} \in B(p, \delta)) \leq \sum_{\nu \in B(p, \delta) \cap \mathcal{P}_n} e^{-nD(\nu\|q)}$$

$$\leq |B(p, \delta) \cap \mathcal{P}_n| \, e^{-n \inf_{\nu \in B(p, \delta)} D(\nu\|q)}$$

$$\leq (n+1)^d \, e^{-n \inf_{\nu \in B(p, \delta), q \in Q} D(\nu\|q)},$$

where the last inequality follows since $|\mathcal{P}_n| \leq (n+1)^d$. Therefore,

$$e_n(\hat{\sigma}_n^A, \varphi^\delta) \leq (n+1)^d \, e^{-n \inf_{\nu \in B(p, \delta), q \in Q} D(\nu\|q)} + \gamma p(\mathcal{P}_{\mathbf{x}^n} \in B(p, \delta)^c).$$

The first term above goes to 0 as $n \to \infty$, since $\inf_{\nu \in B(p, \delta), q \in Q} D(\nu\|q) > 0$. Also, by the weak law of large numbers, we see that $\mathcal{P}_{\mathbf{x}^n}$ converges to $p$ in probability under the null hypothesis $H_0$. Therefore,

$$p(\mathcal{P}_{\mathbf{x}^n} \in B(p, \delta)^c) \to 0.$$

Hence, we conclude that $e_n(\hat{\sigma}_n^A, \varphi^\delta) \to 0$ as $n \to \infty$. Combining this with (B.1) completes the proof of the Lemma. $\qquad\square$

## B.4 Proof of Lemma 4.3

From Lemma 4.2, we have $e_n(\hat{\sigma}_n^A, \hat{\sigma}_n^D) \to 0$ as $n \to \infty$. Since $u_n^A(q^*, \hat{\sigma}_n^D) \geq -c(q^*)$ and since $(\hat{\sigma}_n^A, \hat{\sigma}_n^D)$ is a NE for all $n \geq 1$, it follows that

$$\int c(q)\hat{\sigma}_n^A(dq) \to c(q^*)$$

as $n \to \infty$. Since $(\hat{\sigma}_n^A)_{n \geq 1}$ is a sequence of probability measures on the compact space $Q$, by Prohorov's theorem (see Theorem 1, Section 2 in Chapter 3 of [27]), there exists a weakly convergent subsequence (say $(n_k)_{k \geq 1}$). Let $\mu$ denote the weak limit of $(\hat{\sigma}_{n_k}^A)_{k \geq 1}$. Then, we have,

$$c(q^*) = \lim_{k \to \infty} \int c(q)\hat{\sigma}_{n_k}^A(dq)$$
$$= \int c(q)\mu(dq), \tag{B.2}$$

where the last equality follows from weak convergence.

We now claim that $\mu = \delta_{q^*}$. Suppose not. Then, there exists $\varepsilon > 0$ such that $\mu(B(q^*, \varepsilon)^c) > 0$. By Assumption (A3), for the above $\varepsilon$, there exists a $\delta > 0$ such that $c(q) > c(q^*) + \delta$ whenever $q \in B(q^*, \varepsilon)^c$. Therefore,

$$\int_Q c(q)\mu(dq) = \int_{B(q^*, \varepsilon)} c(q)\mu(dq) + \int_{B(q^*, \varepsilon)^c} c(q)\mu(dq)$$
$$\geq c(q^*)\mu(B(q^*, \varepsilon)) + (c(q^*) + \delta)\mu(B(q^*, \varepsilon)^c)$$
$$= c(q^*) + \delta\mu(B(q^*, \varepsilon)^c)$$
$$> c(q^*),$$

which contradicts (B.2). Therefore, it follows that $\mu(B(q^*, \varepsilon)^c) = 0$ for every $\varepsilon > 0$ and hence $\mu = \delta_{q^*}$. Since $\mu$ is independent of the subsequence $(n_k)_{k \geq 1}$, it follows that the whole sequence $(\hat{\sigma}_n^A)_{n \geq 1}$ converges to $\delta_{q^*}$ (see Lemma 1, Section 3 in Chapter 3 of [27]). This completes the proof of the lemma. $\square$

To prove Lemma 4.4, we need the following lemma, which asserts uniform convergence of integrals of the relative entropy functional w.r.t. the equilibrium strategy of the attacker.

**Lemma B.1.** *Let $(\hat{\sigma}_n^A, \hat{\sigma}_n^D)_{n \geq 1}$ be as in Lemma 4.2. Then,*

$$\sup_{\mu \in M_1(\mathcal{X})} \left| \int D(\mu||q)\hat{\sigma}_n^A(dq) - D(\mu||q^*) \right| \to 0 \text{ as } n \to \infty.$$

*Proof.* Fix $\varepsilon > 0$ and $\mu \in M_1(\mathcal{X})$. Then, using the uniform continuity of the relative entropy function on $M_1(\mathcal{X}) \times Q$, there exists $\delta > 0$ such that

$$\sup_{q \in Q} |D(\mu||q) - D(\mu'||q)| < \varepsilon \text{ for all } \mu' \in B(\mu, \delta).$$

Therefore, for all $\mu' \in B(\mu, \delta)$, we have

$$\left| \int D(\mu'||q)\hat{\sigma}_n^A(dq) - \int D(\mu||q)\hat{\sigma}_n^A(dq) \right| \leq \int |D(\mu'||q) - D(\mu||q)|\hat{\sigma}_n^A(dq)$$
$$\leq \varepsilon, \text{ for all } n \geq 1.$$

Also, using weak convergence of $(\hat{\sigma}_n^A)_{n \geq 1}$ to the point mass at $q^*$, there exists $N_\mu \geq 1$ such that

$$\left| \int D(\mu||q)\hat{\sigma}_n^A(dq) - D(\mu||q^*) \right| \leq \varepsilon \text{ for all } n \geq N_\mu.$$

Note that, the sets $B(\mu, \delta)_{\mu \in Q}$ is an open cover for $M_1(\mathcal{X})$. By compactness of the space $M_1(\mathcal{X})$, extract a finite subcover $B(\mu_i, \delta), 1 \leq i \leq k$. Put $N = \max\{N_{\mu_1}, \ldots, N_{\mu_k}\}$. Then, for all $n \geq N$, we have

$$\left| \int D(\mu||q)\hat{\sigma}_n^A(dq) - D(\mu||q^*) \right| \leq \left| \int D(\mu||q)\hat{\sigma}_n^A(dq) - \int D(\mu_i||q)\hat{\sigma}_n^A(dq) \right| +$$
$$\left| \int D(\mu_i||q)\hat{\sigma}_n^A(dq) - D(\mu_i||q^*) \right| +$$
$$|D(\mu_i||q^*) - D(\mu||q^*)|$$
$$\leq 3\varepsilon,$$

where $\mu_i$ is such that $\mu \in B(\mu_i, \delta)$. The result now follows since $\varepsilon$ and $\mu$ are arbitrary. $\square$

## B.5 Proof of Lemma 4.4

Recall the decision rule $\hat{\varphi}_n$ from Proposition 4.1. Note that if $H_0$ is accepted under $\hat{\varphi}_n$ when the defender observes $\mathbf{x}^n$, then we have

$$\frac{\int q(\mathbf{x}^n)\hat{\sigma}_n^A(dq)}{p(\mathbf{x}^n)} \leq \gamma$$

(note that there could be randomization when equality holds above). By Proposition 4.1, notice that $(\hat{\sigma}_n^A, \hat{\varphi}_n)$ is a Nash equilibrium, and $e_n(q, \hat{\sigma}_n^D) = e_n(q, \hat{\varphi}_n)$ for all $q \in Q$. Therefore it suffices to show that $\sup_{q \in Q} e_n(q, \hat{\varphi}_n) \to 0$ as $n \to \infty$.

Note that, the acceptance region of $H_0$ under the decision rule $\hat{\varphi}_n$ is type-based, i.e., for every $n$-length word $\mathbf{x}^n$, $\hat{\varphi}_n(\mathbf{x}^n)$ depends only on $\mathcal{P}_{\mathbf{x}^n}$. Therefore, if $H_0$ is accepted when the defender observes $\mathbf{x}^n$, the type $\mathcal{P}_{\mathbf{x}^n}$ must belong to the following subset of $M_1(\mathcal{X})$:

$$\left\{ \mathcal{P}_{\mathbf{x}^n} : \log \int \frac{q(\mathbf{x}^n)}{p(\mathbf{x}^n)} \hat{\sigma}_n^A(dq) \leq \log \gamma \right\}.$$

Define

$$\Gamma_n = \left\{ \mathcal{P}_{\mathbf{x}^n} : \int \log \left( \frac{q(\mathbf{x}^n)}{p(\mathbf{x}^n)} \right) \hat{\sigma}_n^A(dq) \leq \log \gamma \right\}.$$

Notice that, by Jensen's inequality, the acceptance region of $H_0$ under the decision rule $\hat{\varphi}_n$ is a subset of the above set $\Gamma_n$. Also, it is easy to check that,

$$\Gamma_n = \{ \mu \in M_1(\mathcal{X}) : D(\mu||p) - \int D(\mu||q)\hat{\sigma}_n^A(dq) \leq \frac{\log \gamma}{n} \} \cap \mathcal{P}_n.$$

We now show that the set $\Gamma_n$ does not intersect the set $Q$ for large enough $n$. First, notice that, the set $\{\mu \in M_1(\mathcal{X}) : D(\mu||p) \leq D(\mu||q^*)\}$ is closed in $M_1(\mathcal{X})$. Therefore, by Assumption (A4), there exists $\eta > 0$ such that $Q^\eta \cap \{\mu \in M_1(\mathcal{X}) : D(\mu||p) \leq D(\mu||q^*)\} = \emptyset$, where $Q^\eta = \{\mu \in M_1(\mathcal{X}) : \inf_{q \in Q} ||\mu - q|| \leq \eta\}$ is the $\eta$-expansion of the set $Q$.

We show that there exists $N \geq 1$ such that $Q^\eta \cap \Gamma_n = \emptyset$ for all $n \geq N$. Suppose not, then we can find a sequence $(\mu_n)_{n \geq 1}$ such that $\mu_n \in Q^\eta$ and $\mu_n \in \Gamma_n$ for all $n \geq 1$. Since $Q^\eta$ is compact, we can find a subsequence $(n_k)_{k \geq 1}$ along which $\mu_n$ converges, and let $\mu = \lim_{k \to \infty} \mu_{n_k} \in Q^\eta$. Since $\mu_n \in \Gamma_n$ for all $n \geq 1$, using Lemma B.1, we see that $\mu$ satisfies $D(\mu||p) \leq D(\mu||q^*)$. This contradicts the fact that $Q^\eta \cap \Gamma_n = \emptyset$, and hence, there exists $N \geq 1$ such that $Q^\eta \cap \Gamma_n = \emptyset$ for all $n \geq N$.

By the law of large numbers, we have

$$\sup_{q \in Q} q(\mathcal{P}_{\mathbf{x}^n} \notin B(q, \eta)) \to 0,$$

and

$$p(\mathcal{P}_{\mathbf{x}^n} \notin \Gamma_n) \to 0$$

as $n \to \infty$. But, notice that

$$\begin{aligned} e_n(q, \hat{\varphi}_n) &\leq q(\mathcal{P}_{\mathbf{x}^n} \in \Gamma_n) + \gamma p(\mathcal{P}_{\mathbf{x}^n} \notin \Gamma_n) \\ &\leq q(\mathcal{P}_{\mathbf{x}^n} \notin B(q, \eta)) + \gamma p(\mathcal{P}_{\mathbf{x}^n} \notin \Gamma_n) \end{aligned}$$

for all $q \in Q$ and $n \geq N$. Therefore,

$$\begin{aligned} \sup_{q \in Q} e_n(q, \hat{\varphi}_n) &\leq \sup_{q \in Q} q(\mathcal{P}_{\mathbf{x}^n} \notin B(q, \eta)) + \gamma p(\mathcal{P}_{\mathbf{x}^n} \notin \Gamma_n) \\ &\to 0 \end{aligned}$$

as $n \to \infty$. $\square$

## B.6 Proof of Lemma 4.5

Fix $\varepsilon > 0$. By Lemma 4.4, there exists $N_\varepsilon$ such that

$$e_n(q_n, \hat{\sigma}_n^D) \leq \varepsilon$$

for all $n \geq N_\varepsilon$. Therefore,

$$u_n^A(q_n, \hat{\sigma}_n^D) \leq \varepsilon - c(q_n)$$

for all $n \geq N_\varepsilon$. However, by playing the pure strategy $q^*$, the attacker utility is

$$u_n^A(q^*, \hat{\sigma}_n^D) \geq -c(q^*)$$

for all $n \geq 1$. Since $(\hat{\sigma}_n^A, \hat{\sigma}_n^D)$ is a Nash equilibrium, and since $q_n \in supp(\hat{\sigma}_n^A)$, we must have $u_n^A(q_n, \hat{\sigma}_n^D) \geq u_n^A(q^*, \hat{\sigma}_n^D)$ for all $n \geq N_\varepsilon$. That is,

$$c(q_n) \leq c(q) + \varepsilon$$

for all $n \geq N_\varepsilon$. Thus, it follows that, $c(q_n) \to c(q^*)$ as $n \to \infty$. Using the definition of $q^*$, we see that $q_n \to q^*$ as $n \to \infty$. $\square$

## B.7 Proof of Theorem 4.1

First, we obtain the asymptotic lower bound. Towards this, we shall consider an equivalent zero-sum game for $\mathcal{G}^B(d,n)$. For $q \in Q$ and $\varphi \in \Phi_n$, define

$$u_n^{eq}(q,\varphi) = \sum_{\mathbf{x}^n}(1 - \varphi(\mathbf{x}^n))q(\mathbf{x}^n) + \gamma \sum_{\mathbf{x}^n}\varphi(\mathbf{x}^n)p(\mathbf{x}^n) - c(q).$$

Observe that, as far as the attacker is concerned, for any $\varphi \in \Phi_n$, maximizing $u_n^A(\cdot,\varphi)$ is the same as maximizing $u_n^{eq}(\cdot,\varphi)$, as the extra term present in $u_n^{eq}$ does not depend on the attacker strategy. Similarly, for any $q \in Q$, maximizing the defender's utility function $u_n^D(q,\cdot)$ is the same as minimizing $u_n^{eq}(q,\cdot)$, as the cost function $c$ does not depend on the defender's strategy. Therefore, we see that $\mathcal{G}^B(d,n)$ is best-response equivalent to a two-player zero sum game (with attacker being first player and defender being second player) with the above utility for the first player. Hence, any equilibrium for the original game is also going to be an equilibrium for the zero-sum equivalent game with utility function $u_n^{eq}$ (see Definition 4 in [12] and the remarks before Theorem 2).

Consider the strategy profile $(q^*, \hat{\sigma}_n^D)$, i.e., the attacker plays the pure strategy $q^*$ and the defender plays the mixed strategy $\hat{\sigma}_n^D$ that comes from the equilibrium. By definition of the Nash equilibrium, and the equivalence of $\mathcal{G}^B(d,n)$ with the above zero-sum game, we have

$$u_n^{eq}(\hat{\sigma}_n^A, \hat{\sigma}_n^D) \geq u_n^{eq}(q^*, \hat{\sigma}_n^D). \tag{B.3}$$

($u_n^{eq}(\hat{\sigma}_n^A, \hat{\sigma}_n^D)$ denotes the utility in mixed extension of the equivalent zero-sum game).

Define the deterministic decision rule $\varphi_n^*$ by

$$\varphi_n^*(\mathbf{x}^n) = \begin{cases} 1, & \text{if } \frac{q^*(\mathbf{x}^n)}{p(\mathbf{x}^n)} \geq \gamma, \\ 0, & \text{otherwise.} \end{cases}$$

It is easy to see that $\varphi_n^*$ minimizes $e_n(q^*, \cdot)$. Writing the probabilities $p(\mathbf{x}^n)$ and $q(\mathbf{x}^n)$ in terms of $\mathcal{P}_{\mathbf{x}^n}$, it is easy to check that, the acceptance region of $\varphi_n^*$ is given by

$$\Gamma_n^* = \{\nu \in M_1(\mathcal{X}) \cap \mathcal{P}_n : D(\nu||q^*) - D(\nu||p) > \frac{\log \gamma}{n}\},$$

i.e., $\varphi_n^*(\mathbf{x}^n) = 0$ whenever $\mathcal{P}_{\mathbf{x}^n} \in \Gamma_n^*$, and $\varphi_n^*(\mathbf{x}^n) = 1$ otherwise. Noting that $u_n^{eq}(q,\varphi) = e_n(q,\varphi) - c(q)$, (B.3) becomes

$$e_n(\hat{\sigma}_n^A, \hat{\sigma}_n^D) \geq \int \sum_{\mathbf{x}^n}((1 - \varphi(\mathbf{x}^n))q^*(\mathbf{x}^n) + \gamma\varphi(\mathbf{x}^n)p(\mathbf{x}^n))\,\hat{\sigma}_n^D(d\varphi) - c(q^*) + \int c(q)\hat{\sigma}_n^A(dq)$$

$$\geq \int \sum_{\mathbf{x}^n}((1 - \varphi(\mathbf{x}^n))q^*(\mathbf{x}^n) + \gamma\varphi(\mathbf{x}^n)p(\mathbf{x}^n))\,\hat{\sigma}_n^D(d\varphi)$$

$$\geq \sum_{\mathbf{x}^n}((1 - \varphi_n^*(\mathbf{x}^n))q^*(\mathbf{x}^n) + \gamma\varphi_n^*(\mathbf{x}^n)p(\mathbf{x}^n)), \tag{B.4}$$

where the second inequality follows from the definition of $q^*$, and the last inequality follows from the optimality of $\varphi_n^*$. The quantitiy in the RHS of the last inequality is the minimum Bayesian error for the following standard binary hypothesis testing problem: under the null hypothesis, each symbol in $\mathbf{x}^n$ is generated independently from $p$, and under the alternate hypothesis, each symbol is generated independently from $q^*$. It is well known that (see, for example, Corollary 3.4.6 in [10]),

$$\liminf_{n \to \infty} \frac{1}{n}\log e_n(q^*, \varphi_n^*) \geq -\Lambda_0^*(0),$$

and hence, from (B.3) and (B.4), it follows that,

$$\liminf_{n \to \infty} \frac{1}{n}\log e_n(\hat{\sigma}_n^A, \hat{\sigma}_n^D) \geq -\Lambda_0^*(0). \tag{B.5}$$

We now proceed to show the upper bound. Define the decision rule $\varphi_n'$ by

$$\varphi_n'(\mathbf{x}^n) = \begin{cases} 1, & \text{if } \frac{q^*(\mathbf{x}^n)}{p(\mathbf{x}^n)} \geq 1, \\ 0, & \text{otherwise.} \end{cases}$$

Similar to the decision rule $\varphi_n^*$, the acceptance region of $\varphi_n'$ can be written as

$$\Gamma' = \{\nu \in M_1(\mathcal{X}) : D(\nu||q^*) - D(\nu||p) > 0\},$$

i.e., $\varphi_n'(\mathbf{x}^n) = 0$ if $\mathcal{P}_{\mathbf{x}^n} \in \Gamma'$, and $\varphi_n'(\mathbf{x}^n) = 1$ otherwise. By the definition of a Nash equilibrium, and noting that $u_n^D(\hat{\sigma}_n^A, \hat{\sigma}_n^D) = -e_n(\hat{\sigma}_n^A, \hat{\sigma}_n^D)$, we have

$$e_n(\hat{\sigma}_n^A, \hat{\sigma}_n^D) \leq e_n(\hat{\sigma}_n^A, \varphi_n'), \tag{B.6}$$

where $(\hat{\sigma}_n^A, \varphi_n')$ denotes the strategy profile where the attacker plays the mixed strategy $\hat{\sigma}_n^A$ that comes form the equilibrium, and the defender plays the pure strategy $\varphi_n'$. We have,

$$e_n(\hat{\sigma}_n^A, \varphi_n') = \int \sum_{\mathbf{x}^n} \left( (1 - \varphi_n'(\mathbf{x}^n)) q(\mathbf{x}^n) + \gamma \varphi_n'(\mathbf{x}^n) p(\mathbf{x}^n) \right) \sigma_n^A(dq)$$

$$= \int q(\mathcal{P}_{\mathbf{x}^n} \in \Gamma') \hat{\sigma}_n^A(dq) + p(\mathcal{P}_{\mathbf{x}^n} \in (\Gamma')^c).$$

Consider the first term. Using the upper bound on the probability of observing a type under a given distribution (Lemma 2.1.9 in [10]), we have

$$q(\mathcal{P}_{\mathbf{x}^n} \in \Gamma') \leq (n+1)^d e^{-n \inf_{\nu \in \Gamma'} D(\nu \| q)}.$$

Fix $\varepsilon > 0$. Since the relative entropy is jointly uniformly continuous on $\Gamma' \times Q$, there exists a $\delta > 0$ such that

$$D(\nu \| q) \geq D(\nu \| q^*) - \varepsilon$$

for all $\nu \in \Gamma'$ whenever $\|q - q^*\|_2 < \delta$. For the above $\delta$, by Lemma 4.5, there exists $N_\delta$ such that $\|q - q^*\|_2 < \delta$ whenever $q \in \text{supp}(\hat{\sigma}_n^A)$ for all $n \geq N_\delta$. Therefore, we see that, for all $n \geq N_\delta$ and $\nu \in \Gamma'$,

$$D(\nu \| q) \geq D(\nu \| q^*) - \varepsilon \text{ for all } q \in \text{supp}(\hat{\sigma}_n^A).$$

Therefore, for all $n \geq N_\delta$, we have

$$q(\mathcal{P}_{\mathbf{x}^n} \in \Gamma') \leq (n+1)^d e^{-n(\inf_{\nu \in \Gamma'} D(\nu \| q^*) - \varepsilon)}$$

for all $q \in \text{supp}(\hat{\sigma}_n^A)$. For the second term, using Lemma 2.1.9 in [10], we have

$$p(\mathcal{P}_{\mathbf{x}^n} \in (\Gamma')^c) \leq e^{-n \inf_{\nu \in (\Gamma')^c} D(\nu \| p)}.$$

It can be easily shown that (for example, see Exercise 3.4.14(b) in [10]), $\inf_{\nu \in \Gamma'} D(\nu \| q^*) = \inf_{\nu \in (\Gamma')^c} D(\nu \| p) = \Lambda_0^*(0)$. Hence, the above implies that

$$\limsup_{n \to \infty} \frac{1}{n} \log e_n(\hat{\sigma}_n^A, \varphi_n') \leq -\Lambda_0^*(0) + \varepsilon.$$

Letting $\varepsilon \to 0$, we get

$$\limsup_{n \to \infty} \frac{1}{n} \log e_n(\hat{\sigma}_n^A, \varphi_n') \leq -\Lambda_0^*(0).$$

Therefore, from (B.6) and the above inequality, we have

$$\limsup_{n \to \infty} \frac{1}{n} \log e_n(\hat{\sigma}_n^A, \hat{\sigma}_n^D) \leq -\Lambda_0^*(0). \tag{B.7}$$

The theorem now follows from (B.5) and (B.7). $\qquad \square$

## B.8  Proof of Lemma A.1

We show sequential compactness of $\Phi_n$. Let $(\varphi_k)_{k \geq 1}$ be a sequence in $\Phi_n$. Let $\mathbf{x}_1^n, \ldots, \mathbf{x}_{2^n}^n$ denote the elements of $\mathcal{X}^n$. Since $\varphi_n(\mathbf{x}^n) \in [0, 1]$ for all $\mathbf{x}^n \in \mathcal{X}^n$, there exists a subsequence $(k_l^{(1)})_{l \geq 1}$ along which $\varphi(\mathbf{x}_1^n)$ converges. We can then extract a further subsequence $(k_l^{(2)})_{l \geq 1}$ of $(k_l^{(1)})_{l \geq 1}$ along which $\varphi(\mathbf{x}_2^n)$ converges. Repeating the above procedure $2^n$ times, we see that, there exists a subsequence $(k_l)_{l \geq 1}$ along which $\varphi(\mathbf{x}^n)$ converges for all $\mathbf{x}^n \in \mathcal{X}^n$. Put

$$\varphi(\mathbf{x}^n) = \lim_{l \to \infty} \varphi_{k_l}(\mathbf{x}^n), \ \mathbf{x}^n \in \mathcal{X}^n.$$

It is clear that $d_n(\varphi_{k_l}, \varphi) \to 0$ as $l \to \infty$, and we have

$$P_n^{FA}(\varphi) = \sum_{\mathbf{x}^n} \varphi(\mathbf{x}^n) p(\mathbf{x}^n)$$

$$= \sum_{\mathbf{x}^n} \lim_{l \to \infty} (\varphi_{k_l}(\mathbf{x}^n)) p(\mathbf{x}^n)$$

$$= \lim_{l \to \infty} \sum_{\mathbf{x}^n} \varphi_{k_l}(\mathbf{x}^n) p(\mathbf{x}^n)$$

$$= \lim_{l \to \infty} P^{FA}(\varphi_{k_l})$$

$$\leq \varepsilon,$$

since $P^{FA}(\varphi_{k_l}) \leq \varepsilon$ for all $l \geq 1$. This shows that the space $\Phi_n$ equipped with the metric $d_n$ is sequentially compact, and hence compact. $\qquad \square$

## B.9 Proof of Proposition A.1

By Lemma A.1, the strategy space of the defender is compact. Also, the strategy space of the attacker is compact under the standard Euclidean topology on $\mathbb{R}^d$. By Lemma A.2, we see that the utility functions of both players are jointly continuous. Therefore, by the Glicksberg fixed point theorem (see, for example, Corollary 2.4. in [26]), there exists a mixed strategy Nash equilibrium for the game $\mathcal{G}(\varepsilon, d, n)$.

We now show the structure of the equilibrium strategy of the defender. Let $(\hat{\sigma}_n^A, \hat{\sigma}_n^D)$ denote a mixed strategy Nash equilibrium of $\mathcal{G}(\varepsilon, d, n)$. By the property of Nash equilibrium, we have that $e_n(\hat{\sigma}_n^A, \varphi) = e_n(\hat{\sigma}_n^A, \hat{\sigma}_n^D)$ for all $\varphi \in supp(\hat{\sigma}_n^D)$. We claim that $P^{FA}(\varphi) = \varepsilon$ for all $\varphi \in supp(\hat{\sigma}_n^D)$. If there exists $\varphi \in supp(\hat{\sigma}_n^D)$ with $P^{FA}(\varphi) < \varepsilon$, then we can find $\mathbf{x}_0^n \in \mathcal{X}^n$ such that $\varphi(\mathbf{x}_0^n) = 0$ and $\delta > 0$ such that the decision rule defined by $\varphi'(\mathbf{x}^n) = \varphi(\mathbf{x}^n)$ for all $\mathbf{x}^n \neq \mathbf{x}_0^n$, and $\varphi'(\mathbf{x}_0^n) = \delta$ has the property that $P_n^{FA}(\varphi') \leq \varepsilon$ and $e_n(\hat{\sigma}_n^A, \varphi') < e_n(\hat{\sigma}_n^A, \hat{\sigma}_n^D)$. This contradicts the fact that, $(\hat{\sigma}_n^A, \hat{\sigma}_n^D)$ is a Nash equilibrium, which proves our claim.

But, note that

$$
\begin{aligned}
e_n(\hat{\sigma}_n^A, \hat{\sigma}_n^D) &= \int \sum_{\mathbf{x}^n} (1 - \varphi(\mathbf{x}^n)) q(\mathbf{x}^n) \hat{\sigma}_n^A(dq) \hat{\sigma}_n^D(d\varphi) \\
&= \sum_{\mathbf{x}^n} \left[ \int (1 - \varphi(\mathbf{x}^n)) q(\mathbf{x}^n) \hat{\sigma}_n^A(dq) \hat{\sigma}_n^D(d\varphi) \right] \\
&= \sum_{\mathbf{x}^n} \left[ \int (1 - \varphi(\mathbf{x}^n)) q_{\hat{\sigma}_n^A}(\mathbf{x}^n) \hat{\sigma}_n^D(d\varphi) \right]
\end{aligned}
$$

where $q_{\hat{\sigma}_n^A} \in M_1(\mathcal{X}^n)$ is given by

$$
q_{\hat{\sigma}_n^A}(\mathbf{x}^n) = \int q(\mathbf{x}^n) \hat{\sigma}_n^A(dq).
$$

That is, when the attacker plays the Nash equilibrium $\hat{\sigma}_n^A$, the defender faces the problem of distinguishing between the two alternatives: (i) $(X_1, \ldots, X_n)$ is generated by i.i.d. $p$, versus (ii) $(X_1, \ldots, X_n)$ is generated by $q_{\hat{\sigma}_n^A}$. By the Neyman-Pearson lemma, we know that there exists a Neyman-Pearson decision rule $\hat{\varphi}_n \in \Phi_n$ with the property that $P^{FA}(\hat{\varphi}_n) = \varepsilon$ and $e_n(\hat{\sigma}_n^A, \cdot)$ is minimized by $\hat{\varphi}_n$ on $\Phi_n$. Since every $\varphi \in supp(\hat{\sigma}_n^D)$ minimizes $e_n(\hat{\sigma}_n^A, \cdot)$, and $P^{FA}(\varphi) = \varepsilon$, and since each $\mathbf{x}^n \in \mathcal{X}^n$ has positive probability of observing under both $H_0$ and $H_1$, an application of the uniqueness part in Neyman-Pearson lemma (see, for example, Section 5.1 in [13]) yields that that $\hat{\sigma}_n^D = \delta_{\hat{\varphi}_n}$. This completes the proof. □

## B.10 Proof of Lemma A.3

Recall the definition of a Neyman-Pearson decision rule. In the binary case, since the comparison of the ratio $\frac{q(\mathbf{x}^n)}{p(\mathbf{x}^n)}$ to a threshold is the same as comparison of the number of 1's in the $n$-length word $\mathbf{x}^n$ to some other threshold, we see that any Neyman-Pearson decision rule $\varphi$ must necessarily be of the following form:

$$
\varphi(\mathbf{x}^n) = \begin{cases} 0, & \text{if } \mathcal{P}_{\mathbf{x}^n}(1) \in \{0, \frac{1}{n}, \ldots, \frac{k}{n}\}, \\ \pi, & \text{if } \mathcal{P}_{\mathbf{x}^n}(1) = \frac{k+1}{n}, \\ 1, & \text{if } \mathcal{P}_{\mathbf{x}^n}(1) \in \{\frac{k+2}{n}, \ldots, 1\}, \end{cases} \tag{B.8}
$$

for some $\pi \in [0, 1]$ and $0 \leq k \leq n$. Here, $\mathcal{P}_{\mathbf{x}^n}(1)$ denotes the fraction of 1's in $\mathbf{x}^n$. The false alarm probability of the above decision rule is

$$
P_n^{FA}(\varphi) = p\left(\mathcal{P}_{\mathbf{x}^n}(1) \in \{0, \frac{1}{n}, \ldots, \frac{k}{n}\}\right) + \pi p\left(\mathcal{P}_{\mathbf{x}^n}(1) = \frac{k+1}{n}\right).
$$

Since every $n$-length word $\mathbf{x}^n$ has positive probability under the distribution $p$, we see that, there exists a unique $k$ and $\pi$ such that $P_n^{FA}(\varphi) = \varepsilon$. Let $\hat{\varphi}_n$ denote the above Neyman-Pearson decision rule. Then, by the Neyman-Pearson lemma (see, for example, Proposition II.D.1 in [25]), we see that,

$$
\hat{\varphi}_n = \arg \max_{\varphi \in \Phi_n} u_n^D(q, \varphi) \text{ for all } q \in Q.
$$

Thus, the defender has a unique strictly dominant strategy. Using the continuity of $c$, and the continuity of the Type II error term in the attacker's strategy, we see that $u_n^A(\cdot, \hat{\varphi}_n)$ is continuous on Q, and hence there exist a maximum. Therefore, letting the attacker play a pure strategy $\hat{q}_n$ that maximizes $u_n^A(\cdot, \hat{\varphi}_n)$ yields a pure strategy Nash equilibrium $(\hat{q}_n, \hat{\varphi}_n)$. □

To prove Lemma A.6, we need the following lemma, which can be proved similar to Lemma B.1.

**Lemma B.2.** *Let* $(\hat{\sigma}_n^A, \hat{\varphi}_n)_{n\geq 1}$ *be as in Lemma A.4. Then,*

$$\sup_{\mu \in M_1(\mathcal{X})} \left| \int D(\mu||q)\hat{\sigma}_n^A(dq) - D(\mu||q^*) \right| \to 0 \text{ as } n \to \infty.$$

## B.11 Proof of Lemma A.6

Let $\gamma_n$ denote the threshold and $\pi$ denote the randomization used in the decision rule $\hat{\varphi}_n$, i.e., $\hat{\varphi}_n$ is of the form

$$\hat{\varphi}_n(\mathbf{x}^n) = \begin{cases} 1, & \text{if} & \frac{q_{\hat{\sigma}_n^A}(\mathbf{x}^n)}{p(\mathbf{x}^n)} > \gamma_n, \\ \pi, & \text{if} & \frac{q_{\hat{\sigma}_n^A}(\mathbf{x}^n)}{p(\mathbf{x}^n)} = \gamma_n, \\ 0, & \text{if} & \frac{q_{\hat{\sigma}_n^A}(\mathbf{x}^n)}{p(\mathbf{x}^n)} < \gamma_n. \end{cases}$$

We first claim that $\limsup_{n\to\infty} \gamma_n \leq 1$. Since $P^{FA}(\hat{\varphi}_n) = \varepsilon$, we have that

$$p\left( \frac{q_{\hat{\sigma}_n^A}(\mathbf{x}^n)}{p(\mathbf{x}^n)} \geq \gamma_n \right) \geq \varepsilon \tag{B.9}$$

But, using the probability of observing an $n$-length word under a distribution (see, for example, Lemma 2.1.6 in [10]), we have

$$q_{\hat{\sigma}_n^A(\mathbf{x}^n)} = \int q(\mathbf{x}^n)\hat{\sigma}_n^A(dq)$$

$$= \int e^{-n(H(\mathcal{P}_{\mathbf{x}^n}) + D(\mathcal{P}_{\mathbf{x}^n}||q))}\hat{\sigma}_n^A(dq),$$

and

$$p(\mathbf{x}^n) = e^{-n(H(\mathcal{P}_{\mathbf{x}^n}) + D(\mathcal{P}_{\mathbf{x}^n}||p))}.$$

Therefore,

$$p\left( \frac{q_{\hat{\sigma}_n^A}(\mathbf{x}^n)}{p(\mathbf{x}^n)} \geq \gamma_n \right) = p\left( \int e^{n(D(\mathcal{P}_{\mathbf{x}^n}||p) - D(\mathcal{P}_{\mathbf{x}^n}||q))} \geq \gamma_n \right)$$

$$\leq p\left( e^{n(D(\mathcal{P}_{\mathbf{x}^n}||p) - \inf_{q\in Q} D(\mathcal{P}_{\mathbf{x}^n}||q))} \geq \gamma_n \right)$$

By Assumption (A1), we can choose $\delta > 0$ such that $D(\mu||p) < \inf_{q\in Q} D(\mu||q)$ for all $\mu \in B(p, \delta)$. Thus,

$$p\left( e^{n(D(\mathcal{P}_{\mathbf{x}^n}||p) - \inf_{q\in Q} D(\mathcal{P}_{\mathbf{x}^n}||q))} \geq \gamma_n \right)$$

$$= p\left( e^{n(D(\mathcal{P}_{\mathbf{x}^n}||p) - \inf_{q\in Q} D(\mathcal{P}_{\mathbf{x}^n}||q))} \geq \gamma_n, \mathcal{P}_{\mathbf{x}^n} \in B(p, \delta) \right)$$

$$+ p\left( e^{n(D(\mathcal{P}_{\mathbf{x}^n}||p) - \inf_{q\in Q} D(\mathcal{P}_{\mathbf{x}^n}||q))} \geq \gamma_n, \mathcal{P}_{\mathbf{x}^n} \notin B(p, \delta) \right)$$

By law of large numbers, $p(\mathcal{P}_{\mathbf{x}^n} \notin B(p, \delta)) \to 0$, and hence the second term above goes to 0 as $n \to \infty$. Suppose that $\limsup_{n\to\infty} \gamma_n > 1$, then there exists a subsequence $(n_k)_{k\geq 1}$ such that $\gamma_{n_k} > 1$ for all $k \geq 1$. Therefore, along this subsequence, the first term above becomes

$$p(e^{n_k(D(\mathcal{P}_{\mathbf{x}^{n_k}}||p) - \inf_{q\in Q} D(\mathcal{P}_{\mathbf{x}^{n_k}}||q))} \geq \gamma_{n_k}, \mathcal{P}_{\mathbf{x}^{n_k}} \in B(p, \delta))$$

$$\leq p(e^{n(D(\mathcal{P}_{\mathbf{x}^{n_k}}||p) - \inf_{q\in Q} D(\mathcal{P}_{\mathbf{x}^{n_k}}||q))} > 1, \mathcal{P}_{\mathbf{x}^{n_k}} \in B(p, \delta))$$

which goes to 0 as $n \to \infty$ by the choice of $\delta$. This implies that, $p\left( \frac{q_{\hat{\sigma}_n^A}(\mathbf{x}^n)}{p(\mathbf{x}^n)} \geq \gamma_n \right) \to 0$ as $n \to \infty$, which contradicts (B.9). Therefore, we must have $\limsup_{n\to\infty} \gamma_n \leq 1$.

We now argue that, for some $\eta > 0$, the acceptance set of $H_0$ under $\hat{\varphi}_n$ does not intersect the set $Q^\eta$. Towards this, consider the set

$$\Gamma_n = \left\{ \mathcal{P}_{\mathbf{x}^n} : \int \log\left( \frac{q(\mathbf{x}^n)}{p(\mathbf{x}^n)} \right) \hat{\sigma}_n^A(dq) \leq \log \gamma_n \right\}.$$

Notice that, by Jensen's inequality, the acceptance region of $H_0$ under the decision rule $\hat{\varphi}_n$ is a subset of the above set $\Gamma_n$. Also, it is easy to check that,

$$\Gamma_n = \{\mu \in M_1(\mathcal{X}) : D(\mu||p) - \int D(\mu||q)\hat{\sigma}_n^A(dq) \leq \frac{\log \gamma_n}{n}\} \cap \mathcal{P}_n.$$

We now show that the set $\Gamma_n$ does not intersect the set $Q$ for large enough $n$. First, notice that, the set $\{\mu \in M_1(\mathcal{X}) : D(\mu||p) \leq D(\mu||q^*)\}$ is closed in $M_1(\mathcal{X})$. Therefore, by Assumption (A4) there exists $\eta > 0$ such that $\{\mu \in M_1(\mathcal{X}) : D(\mu||p) \leq D(\mu||q^*)\} \cap Q^\eta = \emptyset$.

We show that there exists $N \geq 1$ such that $Q^\eta \cap \Gamma_n = \emptyset$ for all $n \geq N$. Suppose not, then we can find a sequence $(\mu_n)_{n \geq 1}$ such that $\mu_n \in Q^\eta$ and $\mu_n \in \Gamma_n$ for all $n \geq 1$. Since $Q^\eta$ is compact, we can find a subsequence $(n_k)_{k \geq 1}$ along which $\mu_n$ converges, and let $\mu = \lim_{k \to \infty} \mu_{n_k} \in Q^\eta$. Since $\mu_n \in \Gamma_n$ for all $n \geq 1$, using Lemma B.2 and the fact that $\limsup_{n \to \infty} \gamma_n = 0$, we see that $\mu$ satisfies $D(\mu||p) \leq D(\mu||q^*)$. This contradicts the fact that $Q^\eta \cap \Gamma_n = \emptyset$, and hence, there exists $N \geq 1$ such that $Q^\eta \cap \Gamma_n = \emptyset$ for all $n \geq N$.

By the law of large numbers, we have

$$\sup_{q \in Q} q(\mathcal{P}_{\mathbf{x}^n} \notin B(q, \eta)) \to 0$$

as $n \to \infty$. But, notice that

$$e_n(q, \hat{\varphi}_n) \leq q(\mathcal{P}_{\mathbf{x}^n} \in \Gamma_n)$$
$$\leq q(\mathcal{P}_{\mathbf{x}^n} \notin B(q, \eta))$$

for all $q \in Q$ and $n \geq N$. Therefore,

$$\sup_{q \in Q} e_n(q, \hat{\varphi}_n) \leq \sup_{q \in Q} q(\mathcal{P}_{\mathbf{x}^n} \notin B(q, \eta)) \to 0$$

as $n \to \infty$. $\qquad\square$

## B.12 Proof of Theorem A.1

We proceed through similar steps as in the proof of Theorem 4.1. To show the lower bound, we let the attacker play the pure strategy $q^*$ instead of the her equilibrium strategy $\hat{\sigma}_n^A$ for all $n \geq 1$. Since $u_n^A(q, \varphi) = e_n(q, \varphi) - c(q)$, and since $(\hat{\sigma}_n^A, \hat{\varphi}_n)$ is a Nash equilibrium for $\mathcal{G}^{NP}(\varepsilon, d, n)$, we see that

$$e_n(\hat{\sigma}_n^A, \hat{\varphi}_n) \geq e_n(q^*, \hat{\varphi}_n) - c(q^*) + \int c(q) \hat{\sigma}_n^A(dq)$$
$$\geq e_n(q^*, \hat{\varphi}_n)$$
$$\geq e_n(q^*, \varphi_n^*)$$

where $\varphi_n^*$ denotes the best $\varepsilon$-level Neyman-Pearson test for distinguishing $p$ from $q^*$ from $n$ independent samples. Here, the second inequality follows from the definition of $q^*$, and the last inequallity follows from the optimality of Neyman-Pearson test $\varphi_n^*$. Hence, using Stein's lemma (see, for example, Lemma 3.4.7 in [10]), we see that

$$\liminf_{n \to \infty} \frac{1}{n} \log e_n(\hat{\sigma}_n^A, \hat{\varphi}_n) \geq -D(p||q^*). \tag{B.10}$$

We now show the upper bound. Fix $0 < \delta < 1$ such that $B(p, \delta) \cap Q = \emptyset$, and consider the deterministic decision rule $\varphi^\delta$ with acceptance region $B(p, \delta)$, i.e., $\varphi^\delta(\mathbf{x}^n) = 0$ whenever $\mathcal{P}_{\mathbf{x}^n} \in B(p, \delta)$ and $\varphi^\delta(\mathbf{x}^n) = 1$ otherwise. To obtain the upper bound, we let the defender play the strategy $\varphi_n^\delta$ for all $n \geq 1$. Since $(\hat{\sigma}_n^A, \hat{\varphi}_n)$ is a Nash equilibrium, and $u_n^D(q, \varphi) = -e_n(q, \varphi)$, we have

$$e_n(\hat{\sigma}_n^A, \hat{\varphi}_n) \leq e_n(\hat{\sigma}_n^A, \varphi^\delta)$$
$$= \int q(\mathcal{P}_{\mathbf{x}^n} \in \Gamma^\delta) \hat{\sigma}_n^A(dq)$$
$$\leq \int (n+1)^d e^{-n \inf_{\nu \in \Gamma^\delta} D(\nu||q)} \hat{\sigma}_n^A(dq), \tag{B.11}$$

where the last inequality follows form the upper bound in Lemma 2.1.9 in [10]. By Lemma A.7 and by the uniform continuity of $D(\cdot||\cdot)$ on $\Gamma^\delta \times Q$, there exists $N_\delta \geq 1$ such that

$$D(\nu||q) \geq D(\nu||q^*) - \delta \text{ for all } \nu \in \Gamma^\delta, q \in supp(\hat{\sigma}_n^A) \text{ and } n \geq N_\delta.$$

Therefore, (B.11) implies that

$$\limsup_{n \to \infty} \frac{1}{n} \log e_n(\hat{\sigma}_n^A, \hat{\varphi}_n) \leq -\inf_{\nu \in \Gamma^\delta} D(\nu||q^*) + \delta.$$

Letting $\delta \to 0$ and using the continuity of $D(\cdot||q^*)$ on $M_1(\mathcal{X})$, we get

$$\limsup_{n \to \infty} \frac{1}{n} \log e_n(\hat{\sigma}_n^A, \hat{\varphi}_n) \leq -D(p||q^*). \tag{B.12}$$

The result now follows form (B.10) and (B.12). $\qquad\square$

(a) $n = 200$                                    (b) $n = 250$

Figure 2: Best response plots for $c(q) = |q - 0.8|$

(a) $c(q) = |q - 0.8|$, $n = 250$, defender plays threshold 166

(b) $c(q) = (q - 0.8)^2$, $n = 800$, defender plays threshold 529

Figure 3: Finer plots of attacker revenue around $q^*$ for specific defender thresholds

## C   Additional Numerical Experiments

### C.1   Bayesian Formulation

As explained in Section 5, we fix $\mathcal{X} = \{0, 1\}$ and $p = 0.5$. For numerical computations, we discretize the set $Q$ into 100 equally spaced points, and we only consider deterministic threshold-based decision rules for the defender.

We first examine the best response of the players. We fix $Q = [0.7, 0.9]$ and the cost function to be $c(q) = |q - q^*|$ where $q^* = 0.8$. Figure 2(a) shows the best response of the players for $n = 200$. The $x$-axis shows the strategy space of the attacker and $y$-axis shows the defender's threshold. The blue curve plots the best response of the defender, and the red curve plots the best response of the attacker for 20 thresholds around the best response threshold corresponding to $q^*$. As we see from the figure, the two curves do not intersect (the best threshold for 0.8 is 133, whereas the best value of $q$ for threshold 133 is 0.7) and hence this suggests that there is no pure strategy equilibrium in this case. Figure 2(b) plots the best response curves for $n = 250$. We see that the two curves intersect (the point of intersection is when the attacker plays 0.8 and the defender plays the threshold 166). However, this does not mean that there is a pure equilibrium, as our discretization may not capture the exact value of the attacker strategy. To see whether this is the case, we plot the attacker revenue when the defender plays the threshold 166 over a finer grid around $q^*$ (1000 equally sized points on the interval of length $1/(100 * n)$ around $q^*$), which is shown in Figure 3(a). From this, we observe that the maximum of the attacker utility is indeed attained at the point $q = 0.8$. This suggests that there is a pure strategy Nash equilibrium when the attacker plays 0.8 and defender plays the threshold 166, though we could not prove this analytically.

Similar to the best response plots in Figure 2, we plot the best response plots for the quadratic cost function $c(q) = (q - q^*)^2$ where $q^* = 0.8$. Figure 4(a) shows the best response plots for $n = 700$. Here, they don't intersect (the best threshold for 0.8 is 463 whereas the best value of $q$ for threshold 463 is 0.7), which shows that there is no pure equilibrium for the game. Figure 4(b) shows the best repose plots for $n = 800$. Here, we see that the curves intersect (when the attacker plays 0.8 and defender plays the threshold 529). As before, Figure 3(b) shows a finer plot of the utility of the attacker around the point $q^*$. We see that the utility of the attacker is indeed maximized at 0.8, which suggests that there is a pure strategy equilibrium when the attacker plays the strategy 0.8 and defender plays the threshold 529.

(a) $n = 700$          (b) $n = 800$

Figure 4: Best response plots for the cost function $c(q) = (q - 0.8)^2$

(a) $c(q) = 2|q - 0.9|$         (b) $c(q) = (q - 0.9)^2$

Figure 5: Error exponents as a function of $n$

From these experiments for the linear as well as quadratic cost functions, as we expect, there is no incentive for the attacker to deviate much form the point $q^*$, since for large values of $n$, the error term in the utility of the attacker does not contribute to the overall revenue. However, in the second case, since the cost function has zero derivative at $q^*$, it is not clear whether a slight deviation form the point $q^*$ can increase the error term compared to the decrease in the cost function, so that the overall utility of the attacker increases. Therefore, the existence of a pure strategy Nash equilibrium with the attacker strategy being equal to $q^*$ is surprising in this case. However, in the first case, since the left and right derivatives of the cost function at $q^*$ are non-zero, the decease in the cost function is much larger compared to the possible increase in the error term as we slightly deviate from $q^*$, and hence it is reasonable to expect the existence of a pure strategy equilibrium at $q^*$ for large $n$.

Comparing the two cost functions, a much higher value of $n$ is needed in the second case for us to have a pure equilibrium at $q^*$, since the increase in the cost function is much slower in the second case as we move away from the point $q^*$.

We now give two more examples that does not satisfy Assumption (A4) whose error exponents are different from Theorem 4.1. As before, $Q = [0.6, 0.9]$ and $q^* = 0.9$, and recall that Assumption (A4) is not satisfied in this case. We consider the linear cost function $c(q) = 2|q - q^*|$ and the quadratic cost function $c(q) = (q - q^*)^2$. Figures 5(a) and 5(b) show the error exponent at the equilibrium as a function of $n$, from $n = 100$ to $n = 400$ in steps of 100. From these plots, we see that, the error exponents converge to somewhere around 0.023 and 0.011 respectively for the above two cost functions, whereas the value of $\Lambda_0^*(0)$ is around 0.111.

## C.2 Neyman-Pearson Formulation

We fix $\varepsilon = 0.1$ and consider the piecewise linear cost function $c(q) = |q - q^*|$ on $Q = [0.7, 0.8]$ where $q^* = 0.8$. As suggested by Lemma A.3, there exists a pure strategy Nash equilibrium for $\mathcal{G}^{NP}(\varepsilon, 2, n)$ for each $n \geq 1$. We first compute the dominant decision rule of the defender by finding the appropriate value of threshold and randomization. Once this is done, we compute the equilibrium by finding the best response of the attacker corresponding to this dominant strategy of the defender (as before, we discretize the set $Q$ into 100 equally-spaced points). We repeat the experiment for different values of $n$, and for the quadratic cost function $c(q) = 0.001 * (q - 0.8)^2$. Figure 6 and Figure 7 shows the results for the above two cost functions.

(a) Attacker's strategy as a function of $n$

(b) Error exponent as a function of $n$

Figure 6: $c(q) = |q - 0.8|$

(a) Attacker's strategy as a function of $n$

(b) Error exponent as a function of $n$

Figure 7: $c(q) = 0.001 * (q - 0.8)^2$

Since the former cost function increases much faster than the latter as we move away from the point $q^*$, we see that the attacker has much more incentive to play a strategy that is away from $q^*$ in the second case compared to the first. This is reflected in the equilibrium strategy of the attacker; from Figures 6(a) and 7(a), we see that it takes much larger values of $n$ for the equilibrium strategy of the attacker to become equal to $q^*$ in the second case compared to the first. From Figures 6(b) and 7(b), we see that the error exponents approach the limiting value $D(p||q^*) = 0.223$.