[Reviews · NeurIPS 2019]

Reviewer 1



Summary of the model: A set of samples is either drawn from p or from some q chosen by an attacker from a set Q. The defended must look at the samples and decide which is the case. The attacker gets utility if the defender decides incorrectly, but pays some cost for drawing the samples that depends on the choice of q. Summary of results: Shows existence of a mixed-strategy Nash equilibrium. Leaves open existence of pure strategy, or natural conditions under which the equilibrium is pure (it seems to me this would be a very nice and likely result, given some strengthening of the assumptions). Shows that in equilibrium, error rates concentrate to zero as the number of samples n grows large. Gives an asymptotic result on the rate of concentration, which is exponential in n. Originality: medium-high. Combines ideas from different literatures to ask an interesting new question at the intersection. Quality: medium. Well-written and organized, results are a strong starting point for the problem introduced. Clarity: medium-high in my opinion. Significance: medium-high in my opinion. Hopefully will spark more work in this are in both modeling and results; perhaps applications as well. Strengths: - the paper is well-written and well-organized. However, I admit that I did not fully understand where the model was going from the introduction, until I got to Section 3 itself. - the model is an interesting point to bridge from. It incorporates aspects of adversarial classification, hypothesis testing, and strategic attacker-defender games (strategic classification). - the results are nice starting points as well, although I feel it seems relatively clear these must be true given the assumption that p and Q are separated. Drawbacks: - not a constructive algorithm or computational results, though this feels possible. - model of attacker feels specific. It makes sense that the separation of Q from p is necessary for these results. But it feels natural that in many cases, the attacker could construct a strategy by first sampling from p, then making arbitrary (perhaps small) changes. Other: Many questions arise for me, but it doesn't mean the paper is incomplete -- these could be future work. Can we compute an equilibrium; is it unique? (If not, are there natural assumptions where it is?) Is it ever pure-strategy; are there assumptions under which it is always pure-strategy? It would also be interesting to understand the "gain" we get from knowing the game is not zero-sum and the adversary is strategic, and playing an equilibrium, versus if we had just played a min-max strategy and assumed the adversary only tried to minimize our utility. For works like this, one may find the "Stackelberg equilibrium" concept useful. It supposes that the defender first commits to a strategy, then the attacker responds. It is used in some of this security games literature and may make sense here as well. That may turn out to have computational or existence implications. Nice job on this paper and please respond to any of my comments if you feel it would be helpful. -------- After response: Thanks for the author response. It was clear, interesting, and informative.

Reviewer 2



The authors consider an adversarial hypothesis testing game. There is a fixed null hypothesis p known to both the attacker and an external agent. The type of the external agent is determined to either be an attacker with probability theta, or non-attacker with probability 1-theta. If the external agent is a non-attacker, the defender observes n examples drawn from p. If the external agent in an attacker, the defender observes n examples drawn from q != p selected by the attacker. A pure strategy of the attacker is a function which classifies its observations as coming from p or not p. The utility of the defender is to minimize a weighted combination of its Type I and Type II errors. The utility of the defender is the probability that samples from q are misidentified as coming from p (i.e., the defender makes a Type II error) minus a cost term for playing q. The cost function for the attacker is assumed to be continuous with a unique minimum q*. The paper also makes the additional assumption that distributions closer to p than q* (in KL) are not available as a pure strategy for the attacker (A4). While some regularity assumption seems necessary, this seems constraining essentially guaranteeing the attacker plays a point near q* in equilibrium, as this both minimizes cost and is difficult for the defender to detect. I wonder if a weaker assumption is possible here. Since the strategy sets belonging to the attacker and defender are uncountable, some work is required to establish the existence of a Nash. Moreover, the authors are able to prove that every Nash equilibrium can be replaced with one where the defender plays a likelihood, thresholding on q(x_n)/p(x_n) for observations x_n. This is an interesting result connecting the setting to ordinary hypothesis testing. Even without A4, the paper shows weak convergence results as n -> infinity. Namely, the defender’s error vanishes in the limit of infinite samples, and the attacker converges to the minimum cost distribution. Finally, the main result of the paper states that the defender’s error vanishes exponentially in the limit, and characterizes the exponent of the decay. The exponent from classical hypothesis testing is recovered. Overall the paper is clear and easy to read, and the results are interesting.

Reviewer 3



I enjoyed reading this submission and would support its publication in NeurIPS. The work studies an adversarial testing situation in which a defender accepts or rejects a null hypothesis based on n independent samples, the null distribution is fixed, and an adversary chooses the alternative distribution. This is a classical setting of minimax testing, except that the adversary has an additional cost associated to picking each alternative (and this cost function is known to the defender). I find the problem and perspective original, and both the paper and the proofs are well-written with attention paid to details. The specific adversarial setting considered in this paper is a bit simple, as the cost c(q) does not vary with n so that the adversary's strategy degenerates (in some sense) in the limit of large n. But I think this paper may lead to interesting follow-up work that considers more nuanced asymptotic settings. A couple comments/questions: 1. Would some weaker version of Assumption (A4) be sufficient in the Neyman-Pearson (not Bayesian) framework? For example, in the 1-dimensional example of Section 5, the optimal test by the defender against the fixed alternative q* in the Bayesian setting might not achieve asymptotically perfect power against a different alternative q. But for the Neyman-Pearson setting, at least intuitively it seems like the acceptance region of an optimal test should be a "vanishingly small neighborhood" of the null distribution p, so I might intuitively expect that the result of Lemma A.6 should always hold. Perhaps there is a different counterexample for the Neyman-Pearson setting in higher dimensions? In any case, I'd appreciate some discussion of this, perhaps deferred to the supplement. 2. Is it necessary to think about a defender's mixed strategy over Phi_n, which is already a space of randomized tests? (This feels like too many levels of mixing...) Is it true that for any mixed strategy sigma_n over Phi_n, the defender can always achieve the same objective with the marginalized pure strategy phi_n(x) = integral phi(x) dsigma_n(phi)? ------------------------ Post-rebuttal: Thanks to the authors for the responses, clarification, and further discussion.

[Author Response · NeurIPS 2019]

We thank the reviewers for their positive feedback and valuable suggestions. Overall, as the reviewers point out, our work is the first analysis of a nonzero-sum adversarial hypothesis testing model bridging multiple areas and is meant to be followed up upon. Our aim was to obtain a tractable game-theoretic model that is general enough to capture some of the problems that arise in adversarial classification yet simple enough to state concrete results on the behavior of any NE and we hope that our work is a first step in that direction. Yet, we greatly appreciate the reviewers suggestions of improvement and further work—below we discuss some of the main questions/comments in the reviews.

**Computation of equilibria:** We have indeed not looked into the problem of computing a NE in our game. We believe that obtaining the complete structure of a NE and computing it is a difficult problem in general because the strategy spaces of both players are uncountable (and there is no pure-strategy NE in general), and we cannot use the standard techniques for finite games; but it is definitely an interesting direction for future developments. We note, however, that in this paper we are able to show a partial structure of NE (the defender performs a certain likelihood ratio test at NE (Proposition 4.1) and the attacker's strategy concentrates (Lemma 4.5)) and that we are able to derive error exponents associated with classification error (at any NE) using a small set of assumptions and without explicitly computing a NE.

**Existence of pure-strategy equilibria:** In the numerical example of Section 5 in one dimension (see Appendix C.1 for the corresponding discussion), it appears that there is no pure-strategy NE for small $n$ but there may be a pure-strategy NE for large $n$ (though we did not prove it). Given the simplicity of this example, we are not sure what kind of natural assumption would lead to obtain existence of a pure strategy NE for all $n$ but investigating such conditions is certainly an interesting direction; and we might be able to show that there exists a pure NE for large-enough $n$.

**Defender's mixed strategy over $\Phi_n$:** In order to show the existence of an equilibrium, it is necessary to consider randomization over $\Phi_n$. However, once the existence of a NE is established, then the test proposed by the reviewer will indeed achieve the same objective- this is shown in Proposition 4.1. If we do not randomize over $\Phi_n$, it is not clear how one can establish existence of an equilibrium, since the objective functions are not quasiconcave—see Remark 4.2.

**Assumption (A4):** As the reviewers pointed out, we agree that this is in fact a strong assumption. This is the condition that naturally appears in the study of error exponents (see line 563 in the proof of Lemma 4.4); and we provide a numerical counter-example where our results do not hold when (A4) is not satisfied. Still, there could be a weaker assumption under which our results hold, but this needs to be further investigated and looking more precisely at what happens when (A4) does not hold is in any case indeed an interesting direction for future work. However, we believe that our work is a good starting point to understand the equilibrium behavior and obtain error exponents for our model.

**Assumption (A4) for the Neyman-Pearson formulation:** We agree with the reviewer that, in one dimension, the acceptance region of an optimal Neyman-Pearson test for a fixed alternative $q$ will be a "vanishingly small neighborhood of the null distribution p" and that while it can still intersect $Q$ for finite $n$, it may not for large-enough $n$; so that Lemma A.6 may always hold. However, it is unclear to us how this might generalize to higher dimension. Our intuition is that in higher dimension, the acceptance region may become close to $p$ only in certain directions. We also note that our proof of Lemma A.6 actually uses Assumption (A4) and not a weaker version of it—see the expression of $\Gamma_n$ in line 690. Overall, we believe that (A4) is needed in higher dimensions even for the Neyman-Pearson case; although it is possible that a weaker assumption will suffice in one dimension—we still need to check that carefully. We will include a discussion about this in the appendix.

**Applications and attacker's model:** Our model is relevant for problems that arise in the context of adversarial classification, as mentioned in Section 3.2 on model discussion; but our focus in this study was indeed more on developing a model that allows analytical investigation while containing the key elements of a nonzero-sum adversarial setting and that could be the basis of further works extending our results and our model. A model where the attacker changes the distribution gradually has been considered in a previous work by Brandão et al. [5]. However, they study a non-game-theoretic setting in the sense that they look for an optimal decision rule in which the adversary can generate each sample from a potentially different distribution (among a given set of distributions). Our nonzero-sum game-theoretic model on the other hand better captures the interaction between rational agents that may arise in some adversarial classification problems. We will look into a possible game-theoretic formulation for the model suggested by the reviewer with a suitable application in the future. We also plan to study a sequential version of our problem where data samples arrive over time and the defender can make a decision in an online fashion.

**Gain from minimax strategy and Stackelberg equilibrium:** We agree with the reviewer that it would be interesting to understand the gain we get from knowing that the game is nonzero-sum and in particular knowing $c(\cdot)$ (note that the gain in utility will be in the exponents). We note, however, that to obtain a completely meaningful comparison, we would need to model the information available to the attacker and the strategy that he adopts, which would lead to a significantly more complex incomplete information game. We also agree that looking at the Stackelberg equilibrium could be interesting and help solve computational issues, although we note that most of the security games literature using Stackelberg games assumes finite action spaces.

We thank the reviewers again for their encouraging feedback and numerous thoughtful suggestions for future extensions.

[Meta-Review · NeurIPS 2019]

The paper proposes a new adversarial framework for hypothesis testing, in a game-theoretic setup. The main positives are: the formulation bridges many fields including statistics, property testing, game-theory, and has the potential to inspire much future work. The theoretical results are reasonable but somewhat unsurprising.